# The emerging role of drought as a regulator of dissolved organic carbon in boreal landscapes

Tejshree Tiwari [1] ✉, Ryan A. Sponseller[2] & Hjalmar Laudon [1]

One likely consequence of global climate change is an increased frequency and intensity of droughts at high latitudes. Here we use a 17-year record from 13 nested boreal streams to examine direct and lagged effects of summer drought on the quantity and quality of dissolved organic carbon (DOC) inputs from catchment soils. Protracted periods of drought reduced DOC concentrations in all catchments but also led to large stream DOC pulses upon rewetting. Concurrent changes in DOC optical properties and chemical character suggest that seasonal drying and rewetting trigger soil processes that alter the forms of carbon supplied to streams. Contrary to expectations, clearest drought effects were observed in larger watersheds, whereas responses were most muted in smaller, peatland-dominated catchments. Collectively, our results indicate that summer drought causes a fundamental shift in the seasonal distribution of DOC concentrations and character, which together operate as primary controls over the ecological and biogeochemical functioning of northern aquatic ecosystems.

Ongoing climate change is expected to result in more extreme weather conditions that give rise to hydrological drought and a greater frequency of drying-rewetting events in terrestrial and aquatic ecosystems[1]. Although the consequences of amplified drying-wetting cycles for soil microbial processes and carbon (C) cycling on land have received much attention[2,3] less is known about how these events are propagated across land-water boundaries to influence aquatic ecosystems and water quality. Further, what we do know about the effects of drought and rewetting on aquatic ecosystems comes largely from research in biomes where such events are historically common[2,4,5]. By comparison, the potential consequences of such hydrological change for aquatic ecosystems and biogeochemical processes in cold, northern biomes remains poorly investigated[6]. Yet, given the vast pools of organic matter that can be hydrologically mobilized in high latitude soils[7], potential increases in drought frequency and intensified drying-rewetting cycles are likely to have pronounced effects on streams and rivers draining boreal and Arctic landscapes.

Northern streams and lakes are typified by high concentrations of terrestrially-derived dissolved organic carbon (DOC) which plays multiple geochemical[8], biogeochemical[9], and ecological roles[10] and is thus an important indicator of water quality. DOC supply from soils influences the transport and bioavailability of heavy metals and anthropogenic organic compounds[11], represents the main energy source for aquatic food webs[9] and promotes the production of harmful byproducts of chlorine disinfection during drinking water sanitization[12]. Further, variation in DOC 'quality', as represented by shifts in the composition of organic compounds and their degree of biological reactivity, can regulate aquatic ecosystem processes, including rates of microbial metabolism[13] and nutrient transformations[14]. Given this variety of important functions, environmental changes that alter the amount and characteristics of DOC supplied to aquatic systems could have wide-ranging consequences for northern aquatic ecosystems.

DOC production and mobility in landscapes are driven by the combination of soil biogeochemical processes and the strength and timing of hydrological connections across terrestrial source areas[15,16], groundwater systems, and stream channels[17]. It is generally recognized that elevated flows promote DOC supply by

[1]Department of Forest Ecology and Management, Swedish University of Agricultural Sciences, SE-901 83 Umea, Sweden. [2]Department of Ecology and Environmental Sciences, Umea Univeristy, 901 87 Umea, Sweden. ✉e-mail: Tejshree.Tiwari@slu.se

strengthening connections between these terrestrial sources and streams[18]. In this way, the timing of high flows, together with temperature-driven changes in soil processes, can shape the overall seasonality of DOC supply to streams[19,20]. However, less is known about how the amount, timing, and chemical character of DOC are altered by seasonal drought episodes, which reduce lateral connectivity, but also set the stage for biogeochemical and microbial processes in dry and disconnected soils[21–23]. Such biogeochemical and microbial changes can alter the pool of organic matter and become mobilized when flow resumes.

Here we ask how the severity of summer drought episodes drives seasonal patterns of DOC quantity and quality in a boreal stream network. To answer this, we investigate how stream and groundwater DOC concentrations, low molecular weight DOC (LMW DOC), carbon/nitrogen (C/N) ratio, and specific UV absorbance at 254 nm (SUVA$_{254}$) respond to summer low flow conditions over 17-years in 13 nested catchments that differ in size and land cover. As the absorbance ratio at A$_{254}$ and A$_{365}$ nm (Abs ratio: A$_{254}$/A$_{365}$) has been found to be negatively correlated with the average molecular weight of organic compounds[24,25], we use this as a proxy for LMW DOC. The C/N ratio provides insight into DOC sources and recalcitrance to degradation and also influences bacterial growth in aquatic systems[26]. Finally, we use SUVA$_{254}$ to assess DOC aromaticity[27,28], where higher values indicate increases in aromatic carbon, which are generally less bioavailable[20,29].

Given the importance of hydrology as a transport vector for DOC, we predict that droughts would disconnect organic-rich soil layers in upper horizons from lateral flow paths, resulting in lower stream DOC, LMW DOC, C/N ratio, and higher SUVA$_{254}$ depending on the severity of the event. During the rewetting phase, we test whether the DOC quantity and quality simply return to pre-drought conditions or whether prolonged dry periods alter the amount and composition of DOC that is mobilized to streams. Finally, we assess how the variation in drying/rewetting shape the DOC response across the drainage network depending on catchment properties, specifically variation in land cover and catchment size.

## Results and discussion
### Mechanisms underlying drought responses

Over the 17-year record, summer low flow hydrology varied considerably, with mean daily minimum discharge ranging several orders of magnitude between the driest and wettest summers (0.0003–0.13 mm day$^{-1}$). The most pronounced summer low flows occurred in 2006 and 2018 with 62 and 41 days below summer discharge of 0.1 mm per day (Fig. 1). This inter-annual variability in hydrology had clear effects on summer DOC concentrations, which declined as drought severity increased. The relationship between drought and DOC held whether or not concentrations were normalized for discharge at the time of sampling (Supplementary Fig. 1), and the two summers with more than 40 days of low flow conditions showed the largest decrease in concentrations from the 17-year average (20–55% lower DOC among the catchments; Fig. 2a). By contrast, the wettest years in the record were associated with elevated summertime DOC concentrations, which increased across the stream network by 10–20% relative to the long-term average. Given the positive relationship between seasonal flow and average concentrations, DOC flux estimates during summer were particularly variable among years, differing within sites by as much as 100%, with the lowest rates coinciding with the driest periods (Supplementary Fig. 2a). Finally, these long-term patterns in stream chemistry were largely mirrored by observations in groundwater, where DOC concentrations also declined as drought severity increased (Supplementary Fig. 3). This relationship was notably strong for riparian soil solution ($r^2 = 0.72$, $p < 0.05$) with drought reducing DOC concentrations in lysimeters by as much as 75% compared with the long-term average. By comparison, DOC

concentrations from wells installed in a headwater mire were relatively unresponsive to drought ($r^2 = 0.11$, $p > 0.1$).

The inter-annual variation in DOC reported here highlights the role of hydrology as a driver of lateral C flux from soils and peatlands to streams[16,18], including the potential for drought to restrict DOC inputs to northern aquatic systems[30], and elsewhere[31,32]. In this case, the riparian forest-stream connection was far more sensitive to drought than was the mire-stream connection, likely reflecting the steep vertical gradient in organic carbon storage for streamside soils, which gives rise to positive relationships between discharge, water table depth, and lateral DOC mobilization[6,33]. Accordingly, the deeper lateral flowpaths, which dominate during protracted low-flow periods, intersect increasingly weaker sources of DOC before reaching streams. While many studies have documented similar positive relationships between DOC concentrations and discharge[34], most emphasize flood responses, and other recent multi-site assessments suggest that this relationship often falls apart under low flow conditions[35]. In this context, the linear declines in concentration with increasing drought severity observed here underscore the acute vulnerability of boreal forest streams to potential hydrological change. Indeed, while a warming climate has the potential to increase DOC inputs to northern streams through enhanced production in soils[36], and severe drought events could reduce DOC solubility in peat soils through geochemical

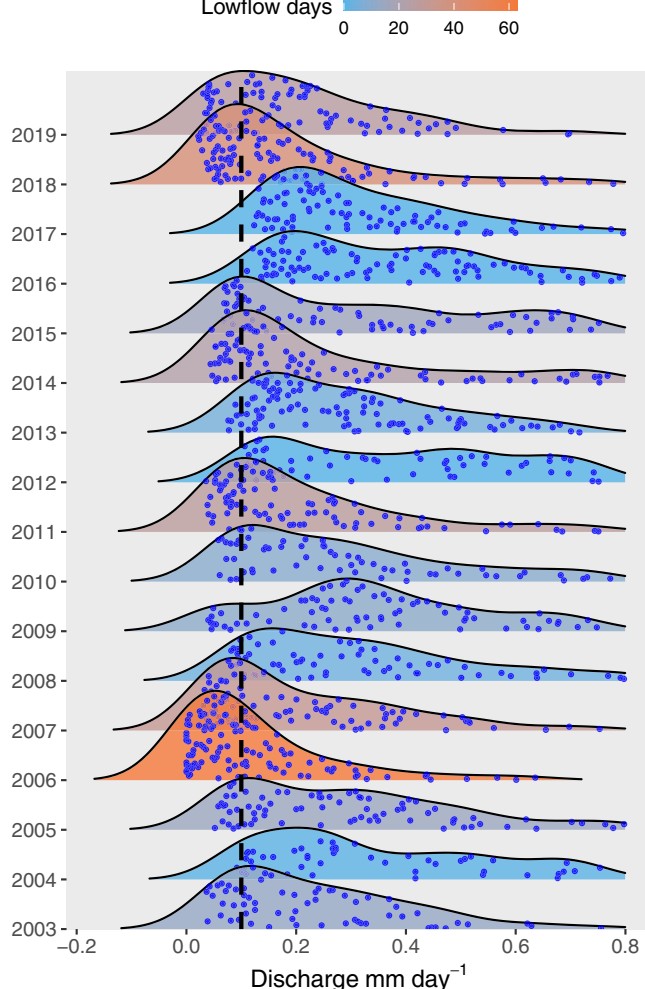

**Fig. 1 | Inter-annual variation in summer discharge in relation to the 0.1 mm per day low flow threshold.** The jitter dots are actual average daily discharge values from site C7 in the Krycklan catchment from which the ridgeline distribution curves were estimated for each summer between 2003 and 2019.

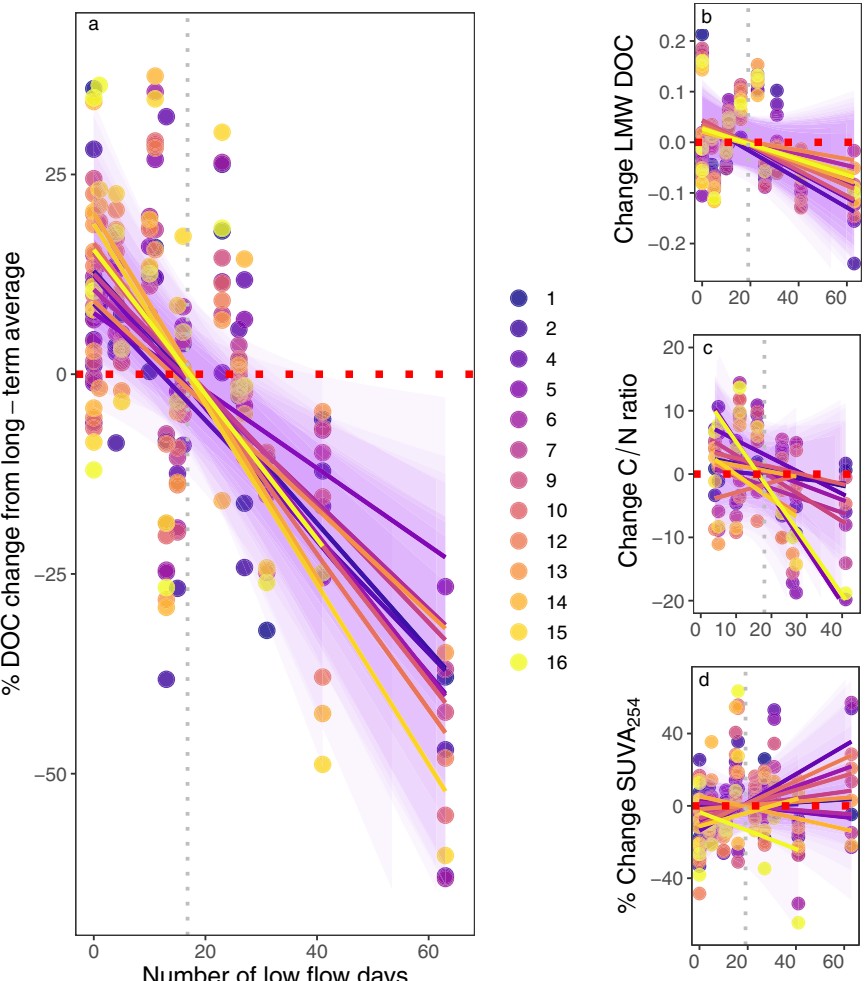

**Fig. 2 | Stream chemical responses to summer drought severity across a boreal stream network. a** percent change in dissolved organic carbon (DOC) ($r^2$ range 0.30–0.65, $p < 0.05$), **b** low molecular weight DOC (LMW DOC) ($r^2$ range 0.20–0.54, $p < 0.05$), **c** carbon to nitrogen ratio (C/N ratio) ($r^2$ range 0.20–0.47 $p < 0.05$), and **d** percent change in specific UV absorbance at 254 nm (SUVA$_{254}$) ($r^2$ range 0.24–0.56, $p < 0.05$), all plotted against the number of low flow days. Points represent the changes for each summer relative to the long-term average, and regression lines are fitted to the data for individual catchments. The red horizontal line indicates zero change while the vertical gray line indicates the average number of low flow days (18 days). Note differences in *y*-axis scales.

mechanisms[19], our results are consistent with the view that any future changes in hydrology, including greater drought frequency, will operate as the first-order control over changes in DOC supply to boreal aquatic ecosystems during summer.

Prolonged dry periods also directly influenced DOC character in streams, although these effects were more variable and in some cases subtle. For example, LMW DOC changed only marginally during drought ($p < 0.05$ for 9 out of 13 sites, Table 1), with declining responses across catchments (Fig. 2b), suggesting that the direct effects of drying were systematic for DOC properties as represented by this particular index. Similarly, the C/N ratio in streams declined consistently at the majority of sites as drought severity increased (Fig. 2c and Table 1, $p < 0.05$ for 12 out of 13 sites). This response most likely reflects well-documented declines in soil C/N with depth in northern forests[37] and mires[38], which results from long-term organic matter processing and potentially the accumulation of microbial N[39]. Thus, as the water table drops during extreme low flow periods, deeper soil strata characterized by more highly processed, lower C/N organic matter emerge as the sole source of dissolved organic matter to streams. More highly processed carbon supplied to streams during dry periods is also supported by the higher SUVA$_{254}$ in the majority of catchments (Fig. 2d), indicating a transition to more aromatic DOC in surface waters. Here, the mire-dominated catchment showed the most

pronounced changes with 50% higher SUVA$_{254}$ (Fig. 2d) during low flow periods, whereas these changes were marginal in the forested dominated sites (Fig. 2d). Altogether, trends in these quality indices were mirrored by observations in the riparian and mire wells, with the largest changes observed in years with the most prolonged summer droughts (Supplementary Fig. 3c, e, g). The observed reduction in LMW DOC and C/N ratio with increases in SUVA$_{254}$ suggest that droughts not only limit the mobility of organic carbon across landscape types, but also reduce the quantity and biological reactivity of DOC across the stream network.

These observed changes in DOC quantity and quality are part of a broader set of physical and chemical responses to low flow conditions in boreal streams that can influence aquatic communities and ecosystem processes during a biologically important time of the year. For headwater environments (e.g., 1st and 2nd order streams), low flow responses can also include widespread hypoxic conditions and accumulation of reduced inorganic solutes[6], which likely have first-order influences on aquatic organisms and community structure[40]. By comparison, for streams, rivers, and lakes embedded further down in the aquatic network, summer DOC declines are likely to have more direct ecological and biogeochemical impacts, given the role of DOC as an energy source to aquatic heterotrophs[41], as a contributor to watercolor that restricts aquatic photosynthesis[10] and mediator of the nutritional

**Table 1 | The regression coefficient of drought and post-drought responses of DOC, LMW DOC, C/N ratio, and SUVA$_{254}$ in the sub-catchments used in this study**

| Sites | Drought | | | | Post-drought | | | |
|---|---|---|---|---|---|---|---|---|
| | DOC | LMW DOC | C/N ratio | SUVA$_{254}$ | DOC | LMW DOC | C/N ratio | SUVA$_{254}$ |
| C1 | 0.42 | 0.20 | 0.25 | 0.02 | 0.33 | 0.24 | 0.14[a] | 0.08[a] |
| C2 | 0.29 | 0.37 | 0.27 | 0.00[a] | 0.39 | 0.56 | 0.5 | 0.01[a] |
| C4 | 0.41 | 0.45 | 0.43 | 0.28 | 0.65 | 0.64 | 0.44 | 0.42 |
| C5 | 0.33 | 0.28 | 0.47 | 0.02[a] | 0.07[a] | 0.36 | 0.58 | 0.02[a] |
| C6 | 0.36 | 0.02[a] | 0.31 | 0.13[a] | 0.42 | 0.36 | 0.20 | 0.03[a] |
| C7 | 0.51 | 0.14[a] | 0.39 | 0.03[a] | 0.67 | 0.27 | 0.40 | 0.31 |
| C9 | 0.43 | 0.20 | 0.27 | 0.02[a] | 0.58 | 0.17[a] | 0.36 | 0.00[a] |
| C10 | 0.52 | 0.04[a] | 0.20 | 0.08[a] | 0.60 | 0.21 | 0.43 | 0.00[a] |
| C12 | 0.65 | 0.02[a] | 0.37 | 0.17[a] | 0.45 | 0.45 | 0.59 | 0.07[a] |
| C13 | 0.3 | 0.01[a] | 0.29 | 0.02[a] | 0.12[a] | 0.59 | 0.14[a] | 0.18[a] |
| C14 | 0.6 | 0.40 | 0.41 | 0.04[a] | 0.28 | 0.31 | 0.46 | 0.23 |
| C15 | 0.52 | 0.54 | 0.13[a] | 0.56 | 0.20 | 0.35 | 0.33 | 0.07[a] |
| C16 | 0.39 | 0.50 | 0.30 | 0.24 | 0.31 | 0.39 | 0.31 | 0.08[a] |

[a]All values significant at $p < 0.05$ except.

quality of algal resources to consumers[42]. Indeed, average summer DOC concentrations for the largest stream (C16) varied by nearly three-fold over the 17 years (5.2–14.6 mg L$^{-1}$), a range comparable to that which is thought to shape aquatic metabolic patterns across broad regional gradients in the northern hemisphere[43]. Thus, by regulating lateral DOC inputs, inter-annual differences in summer hydrology likely play an unappreciated role in driving ecosystem processes of larger boreal streams, rivers, and lakes, including year-to-year shifts in the relative importance of allochthonous versus autochthonous energy sources to aquatic food webs.

## Post-drought recovery of stream biogeochemistry to summer drought

Periods of drought that aerate riparian and wetland soils, while also reducing lateral connectivity, can promote DOC production through decomposition and/or accumulation in upper soil horizons, which can then be mobilized when dry periods are terminated[15,44]. Consistent with this, rewetting following summer drought was associated with elevated DOC flux and concentrations across the stream network, with the largest increases in concentration (100–150%) observed when rewetting followed the most severe dry periods (Fig. 3a, $p < 0.05$, Table 1). Together with the strong declines in concentration during drought, these rewetting responses resulted in a fundamentally different seasonality of DOC supply to streams during drier-than-normal years (Fig. 4). Further, such episodes of DOC flushing to streams were mirrored by increases in concentration in riparian lysimeters (by 50%) and mire wells (by 150%) following prolonged dry periods ($r^2 = 0.31$, $p < 0.05$ and $r^2 = 0.79$, $p < 0.05$, respectively; Supplementary Fig. 3b). Similar to effects on concentration, DOC properties in streams were also influenced by the severity of the preceding drought during the rewetting phase. Specifically, the LMW DOC increased linearly in 12 of the 13 catchments ($p < 0.05$) in relation to prolonged dry periods, while the C/N ratio increased relative to pre-drought conditions in 11 of 13 sites (Fig. 3b, c and Table 1). By contrast, SUVA$_{254}$ decreased during rewetting, with the most prominent changes observed in the forest streams (C1-C2), but this signal also persisted downstream (Fig. 3d). Observations from groundwater reflected similar changes in DOC character during rewetting periods in riparian and mire wells ($p < 0.1$; Supplementary Fig. 3d, f, h). Thus, the changes in both DOC quantity and quality parameters suggest that solutes accumulated/produced during seasonal low flow periods become mobilized during the rewetting period to an extent that is proportional to antecedent drought severity.

Elevated DOC concentrations following the drought periods are in line with the reconnection of near-surface organic-rich soil horizons to adjacent aquatic systems, and essentially represent a temporal redistribution of this land-water exchange. Yet, the increases in DOC concentration, as well as changes in DOM properties, as summer droughts became more severe also suggest that biogeochemical processes influenced soil DOC pools during dry periods. In this context, several mechanisms have been linked to small-scale increases in soil organic matter mineralization and DOC production in response to drying/rewetting cycles[45,46]. For instance, droughts have been found to decrease phenolic microbial inhibitor compounds in wetlands resulting in increased organic matter decomposition and an increase in carbon loss in peats[22,23]. In addition, droughts increase the temperature and degree of aeration of soils that are normally inundated, upregulating organic matter decomposition[31,47]. Finally, rewetting these exposed soils can also trigger the physical and microbial processes that promote rapid organic matter mineralization[2].

While we cannot resolve amongst these mechanisms, the observed patterns suggest that processes in seasonally-exposed organic soils support large pulses of DOC upon rewetting, including a greater fraction of LMW organic forms[29] that are less aromatic[48]. Collectively, these changes in DOC character can directly influence aquatic ecosystems by providing higher-quality organic substrates for heterotrophic bacteria[49]. In fact, based on prior studies in these catchments[50], the observed changes in LMW DOC for surface and groundwater (an increase of ca. 0.2–0.5 units in the Abs ratio) could correspond to as much as a 50% increase in microbial growth efficiency in receiving streams. Consequently, elevated energy mobilization through increased autumn pulses of high-quality DOC is likely to promote aquatic ecosystem respiration[51,52]; indeed, particularly large DOC pulses following drought may create acute periods of anoxia in lakes that cause fish mortality[53]. In addition to these direct biological effects, such DOC pulses may also promote inputs of pollutants to aquatic systems, including a variety of toxic metals that form complexes with organic matter[8,11]. Finally, elevated DOC can cause problems for drinking water treatment via chlorination as reactions with DOC can potentially result in carcinogenic and mutagenic compounds[54]. Together, when combined with longer-term browning trends[55], strong seasonal redistributions of DOC inputs in response to drought may contribute to overall poorer water quality and higher water treatment costs.

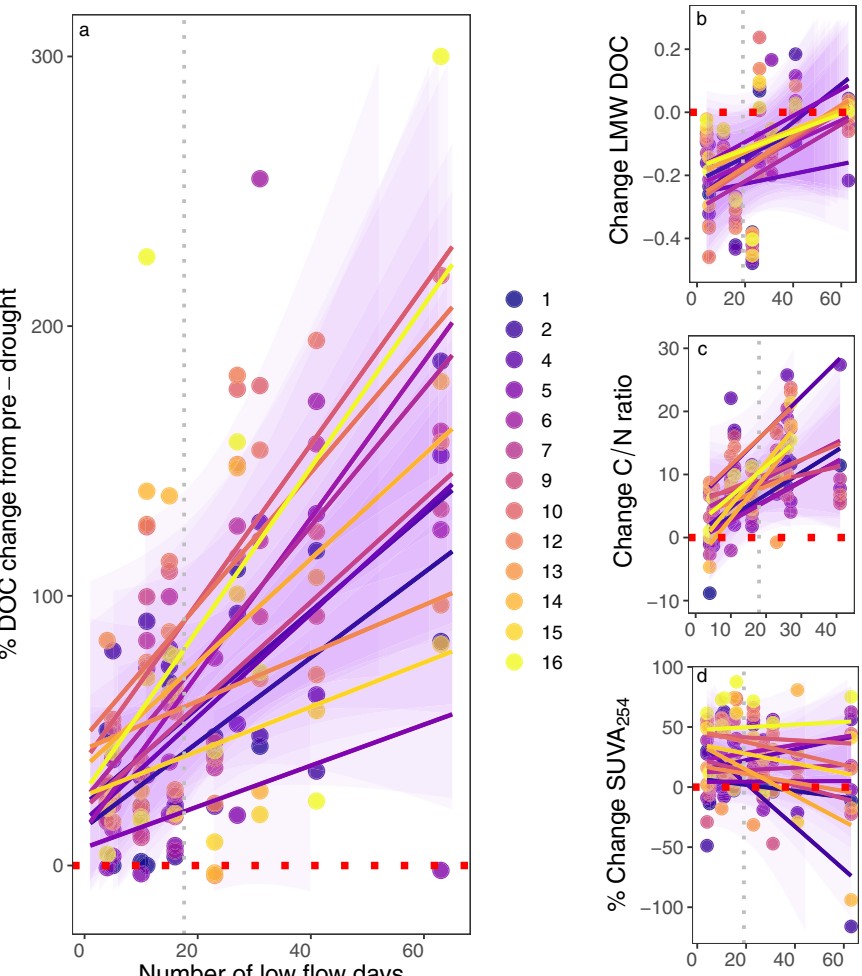

**Fig. 3 | Effects of rewetting in relation to antecedent summer drought in a boreal stream network. a** percent change in dissolved organic carbon DOC ($r^2$ range 0.20–0.67, $p < 0.05$), **b** low molecular weight DOC (LMW DOC) ($r^2$ range 0.21–0.64, $p < 0.05$), **c** carbon to nitrogen ratio (C/N ratio) ($r^2$ range 0.2–0.59, $p < 0.05$), **d** percent change in specific UV absorbance at 254 nm SUVA$_{254}$ ($r^2$ range 0.21–0.64, $p < 0.05$ respectively), all plotted against the number of low flow days in the previous summer. Points represent the changes for each summer relative to the pre-drought (June) averages, and regression lines are fitted to the data for individual catchments. Data for the C/N ratio for the driest summer (2006) are missing because sampling for nitrogen started in 2007. Additionally, for some of the larger sites (C10, C12, C14, C15), sampling stopped in 2017, hence data for 2018 and 2019 were unavailable.

## Network scale responses

Stream DOC responses during drought and post-drought were variable across the river network, reflecting inherent differences in the sensitivity of the sub-catchments to extremely low flows. The variation in these responses was best captured by the difference in catchment size rather than landscape characteristics. Further, the importance of catchment size as a mediator of drought responses differed depending on whether the effects were immediate or lagged. For example, larger catchments showed both the greatest decline in DOC concentration during drought (Supplementary Fig. 4a) and the largest increases in DOC responses upon post-drought rewetting (Supplementary Fig. 4b). Stronger responses to drought in the larger catchments likely relate to their greater distance to near-surface organic DOC sources that feed headwaters (Supplementary Fig. 4). Isolation from these sources is exacerbated by the increasing influence of deeper and DOC-poor groundwater as catchment size increases[17]. As a result, even small losses in lateral and longitudinal connectivity to the more DOC-rich headwaters during drought may cause the chemistry of larger rivers to shift abruptly towards the character of deeper groundwater sources. Upon rewetting, these larger streams and rivers have such low concentrations that the sudden reconnection to upstream DOC sources creates a strong biogeochemical response (Supplementary Fig. 4b, $p < 0.05$ for 10 of the 13 catchments, Table 1). By comparison,

headwaters are seldom supported by these deeper groundwater sources[56], and hence their responses to drying and rewetting events are more attenuated. In this sense, although larger river systems are less prone to complete water loss than headwaters during drought, they nonetheless may show stronger biogeochemical responses to drought events and recovery.

Additionally, while catchment size played an important role in regulating stream chemistry following prolonged dry periods, this effect can be difficult to separate from that of land cover. For instance, the stronger responses by larger, forest-dominated catchments could also be an indication of the degree to which they dry during more severe droughts, as reduced mire cover (and thus greater forest cover) is linked to lower water storage capacity and larger evaporative losses, and thus potentially weakened ecohydrological resilience to drought[57]. Contrastingly, several small catchments are dominated by peat-forming mires and displayed the weakest response during droughts (Supplementary Fig. 4b) and post-drought recovery. This observation suggests that peat-forming wetlands confer resilience to such events, likely by acting as important water storage zones[58] with the potential to dampen the effects of long-term environmental change[59]. As these are all nested catchments, it is not possible to isolate the effects of a single landscape element, still, we expect that the responses observed at the outlet of the catchments are likely buffered by the proportion of

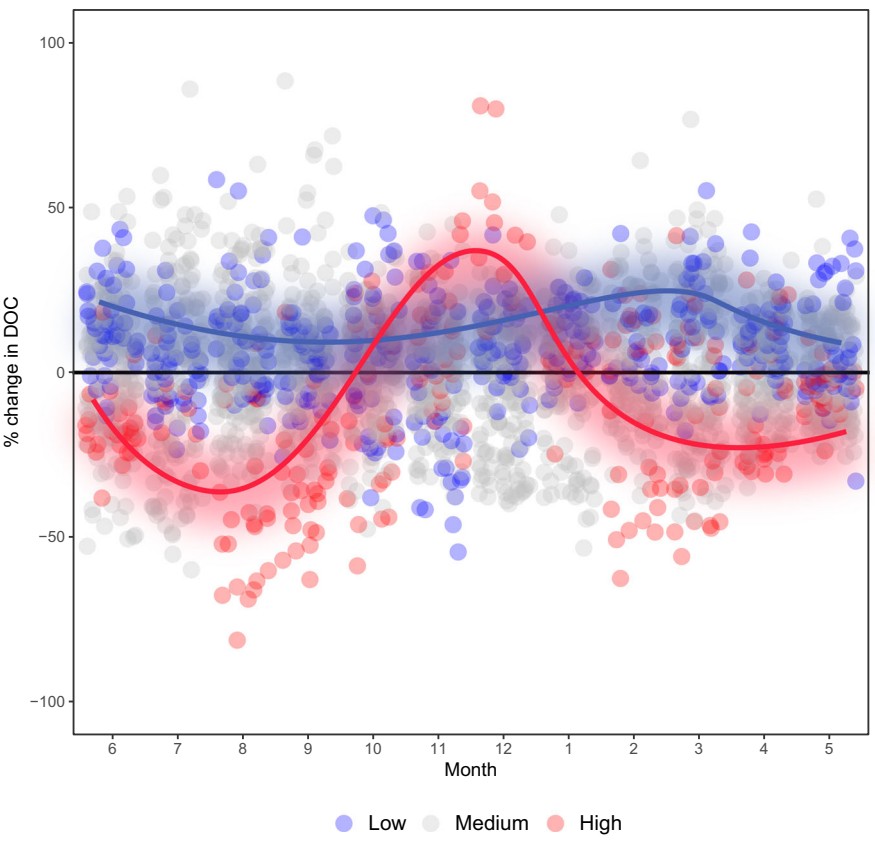

**Fig. 4 | Monthly variation in DOC from 2003 to 2019 expressed as a percentage of long-term averages for the Krycklan sub-catchment during wet, dry, and normal hydrological conditions.** Shown are the long-term dissolved organic carbon DOC averages (black line), the years with high numbers of summer low flow days (above the 90th percentile) (red dots), years with the lowest number of low flow days (blue dots), and the years with the average number of low flow days (gray dots) in Krycklan. The loess regression curves show the average DOC change in years with the highest number of low flow days (red line) and the lowest number of low flow days (blue line).

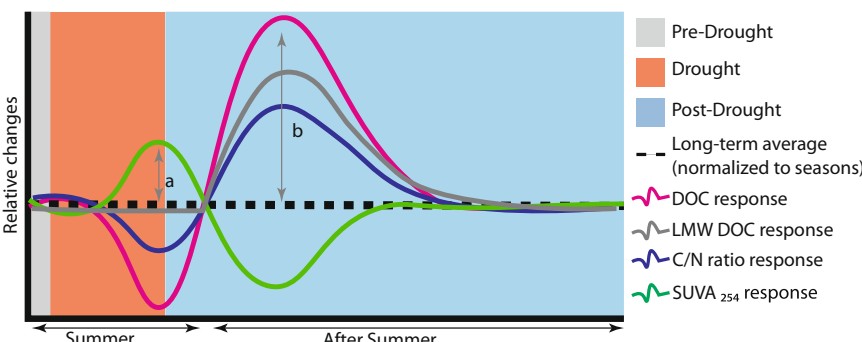

**Fig. 5 | Conceptual responses of dissolved organic carbon (DOC), low molecular weight DOC (LMW DOC), carbon to nitrogen ratio (C/N ratio), and specific UV absorbance at 254 nm (SUVA$_{254}$) to summer low flow conditions.** Different time intervals represent our predictions for **a** drought effects during summer and **b** the post-drought effects after the first rewetting. In all cases, the curves represent predicted changes in relation to long-term averages normalized to seasons.

wetland coverage. However, extended periods of droughts have been suggested to marginalize the resilience of peat and its associated biogeochemical processes[32] and so the patterns observed here could change if drying/wetting cycles are greatly amplified.

## Conceptualizing drought impacts on stream chemistry
The perspectives gained from long-term catchment monitoring help us understand how climate extremes may alter the mobilization of soil organic carbon across spatial and temporal scales in boreal catchments. Observed seasonal variation in amplitude of DOC, LMW DOC,

C/N ratio, and SUVA$_{254}$ suggests that, while drought effects on stream biogeochemistry are direct and immediate, important lagged effects extend beyond the duration of the direct disturbance and are observable across the aquatic network (Fig. 5). Overall, increases in the intensity of drying/rewetting cycles have the potential to shift the seasonality of DOC in boreal streams by reducing summer peaks in concentration (Fig. 5a) while causing anomalously high concentrations during periods of hydrological reconnection later in the autumn (Fig. 5b). Therefore, how recipient aquatic ecosystems cope with less organic energy supplied during the summer season, as well as

potentially large pulses of reactive DOC following droughts in autumn, remain key questions. Further, while much emphasis is currently placed on the direct effects of climate warming at high latitudes[60,61] our study indicates that potential hydrological changes will likely be a more important driver of carbon mobilization and water chemistry change in this region. In this context, current climate projections indicate a warmer and wetter future in Scandinavia, with higher temperatures, earlier snowmelt, and greater evapotranspiration in summer[1]. While understanding how such changes will interact to shape the water balance requires careful study, these climate scenarios point toward the risk of increased drought frequency in the region[62]. Based on our results, such hydrological changes will alter the fundamental properties of boreal aquatic systems as reflected in their seasonal regime of terrestrial DOC supply.

## Methods

### Study area

The Krycklan Catchment Study (KCS) (64.23°N, 19.77°E) is located in Northern Sweden and consists of 13 long-term monitoring streams. The nested sub-catchments vary in size from small headwaters (0.12 km²) to the large outlet (67 km²)[63]. Land cover is dominated by forest till soils (47 to 100% among monitoring sites), lakes (0–6%), and peatlands (referred to as mires) (0–51% areal coverage). Fluvial sediments dominate the lower parts of the catchment below the highest postglacial coastline (Supplementary Table 1). The bedrock consists of 94% metasediments/metagraywacke, 4% acid and intermediate metavolcanic rocks, and 3% basic metavolcanic rocks. Soil mineralogy is dominated by quartz (31–43%), plagioclase (20–25%), K-feldspar (16–33%), amphiboles (7–21%), muscovite (2–16%), and chlorite (1–4%)[64]. Forests are predominantly Scots pine (Pinus sylvestris, 66%), and Norway spruce (Picea abies, 25%) with 9% deciduous forest. Mean annual precipitation recorded between 1991 and 2010 was 610 mm of which 35% was classified as snow during winter (December–April)[65]. In January, the average air temperature is $-9.5 \pm 4.1\,°C$ while July temperatures are $14.5 \pm 1.7\,°C$[65]. In mid-April snowmelt accounts for ~40% of the annual runoff. There are low impacts from land use, with only 2% covered by agricultural lands, <100 inhabitants, and only 0.63% of the catchment is subject to final felling, annually (from 1999 to 2010). The hydrological regime, landscapes, and land uses of Krycklan Catchment are considered representative of the Fennoscandian boreal landscape[66].

### Discharge data

Discharge measurements used to classify summer low flow conditions were based on a small, centrally located headwater sub-catchment (C7, 0.45 km²) for which we have the longest, most detailed record in the Krycklan Catchment. The C7 sub-catchment drains a mix of forest (81%) and mire (19%) land cover, which has a mean specific runoff that falls near the average for all streams in the Krycklan Catchment, and hence provides a reasonable proxy for discharge including drought condition in the area[67]. Using this standardized runoff also made it possible to compare the responses between catchments to the same dry period. Stage height was determined from a 90°V-notch weir in a heated house with a pressure transducer connected to a Campbell Scientific data logger[66]. Daily discharge was calculated from measurements of stage height and an established rating curve based on more than 1000 manual salt dilution and bucket-method measurements[68]. For the flux estimation, discharge was determined for the individual catchment using the same technique used at C7.

### Surface and groundwater sampling

During summer (July–August), surface water samples were collected every other week from each site in acid-washed, high-density polyethylene bottles, which were kept in cold storage until analysis. Samples were collected monthly in winter. Sampling was synchronous across all sub-catchments occurring mostly on the same day, which provided 190 monthly DOC averages during the investigated period (2003–2019) for each of the 13 catchments (Unity Svartberget Data, https://franklin.vfp.slu.se/). In addition, groundwater was sampled from a nest of suction lysimeters that collect forest riparian soil water from 0.10 to 0.65 m at ~ 0.10 m intervals. Finally, mire groundwater samples were collected from wells installed from 2 to 2.5 m. Both sampling of forest riparian soil solution and mire groundwater occurred seasonally; here we used samples collected at the onset of summer (end of May–June) and mid-summer/early autumn (Jul–September). The averages of all depths were used for the analysis of both the riparian and mire wells. Annual sampling in the riparian zone began in 2003. Similar sampling in the mire started later (2009) with sufficient seasonal samples for our purposes, collected on average every second year. Owing to the variation in the sampling regime in the surface water and groundwater, we define pre-summer (end of May–June average), summer (July–August), and post-summer (September) for the drought and post-drought analysis.

In the laboratory, surface water (total 7060 samples from all sites) and groundwater samples (827 samples from both riparian and mire wells) were analyzed for DOC and total nitrogen (TN) concentrations using a Shimadzu TOC-VCPH analyzer after acidification to remove inorganic compounds[59]. Filtered water samples were also analyzed for absorbance using wavelengths ranging from 200 to 600 nm, at 1 nm intervals at a scan speed of 240 nm min⁻¹ and a slit width of 2 nm using a Lambda 40 UV-visible spectrophotometer (Perkin Elmer, Waltham, MA, USA). A 1-cm quartz cuvette with Milli-Q water as the blank was used for measuring the samples. We used the absorbance wavelengths $A_{254}$ and $A_{365}$ nm for the ratio (Abs ratio, $A_{254}/A_{365}$) in this analysis to trace the effects of droughts on the low molecular weight DOC compounds (LMW DOC). The absorption ratio $A_{254}/A_{365}$ is positively correlated to bacterial production in natural waters[24] and negatively to the molecular weight of DOC[25] and thus can be used as a qualitative proxy for LMW DOC. The C/N ratio was calculated by dividing DOC by TN. We did not account for the proportion of dissolved inorganic carbon ($NH_4$, $NO_2$, $NO_3$) in the TN estimates due to the limited inorganic nitrogen data in the larger sites. However, since the proportion of inorganic $N$ is relatively small (average = 9% of the TN during the summer period), we do not expect that this influences our results. To evaluate this, we tested whether the proportion of inorganic $N$ could explain the large percentage change in C/N observed in the driest year (2018). Briefly, we expected that if the proportion of inorganic $N$ was driving changes observed in the C/N ratio, then a higher percentage of inorganic $N$ would correlate with a greater percentage change in the C/N ratio during drought. Instead, we observed the opposite: sites with the greatest proportion of inorganic N during summer had the lowest C/N ratio changes in response to drought. This suggests that variable contributions of inorganic $N$ cannot explain the higher changes in C/N observed across the catchments. Specific UV absorbance ($SUVA_{254}$) was calculated by dividing the UV absorbance at $A_{254}$ nm, measured in inverse meters (m⁻¹), by the DOC concentration[27,28].

### Hydrological droughts

We represented inter-annual variation in the extent of summer drought over the 17 years study period using the occurrence of low flow conditions as a proxy. Within this study period, a threshold of 0.1 mm per day was used to represent low flow conditions, which corresponds to daily discharge less than the 10th percentile value based on summer observations over the last 30 years. The ggridges density distribution function from the ggplot 2 package in R was then used to display the proportion of discharge below the thresholds in each year and to visualize changes in the distribution over space and time. Ridge line plots calculate density estimates from actual data (jitter point, Fig. 1) and plot those using ridgeline visualization. From the density distribution of all the years, we can observe that the

majority of days during the driest summers (2006, 2018) had discharge levels below the 10th percentile (0.1 mm per day; Fig. 1). Years with no low flow days below the threshold were 2012, 2016, and 2017 which showed an almost even distribution of discharge over the summer season (Fig. 1). The years with the average number of low flow days were 2005 and 2015 (15 and 16 days respectively, Fig. 1).

### Drought impacts on DOC, LMW DOC, C/N ratio, and SUVA$_{254}$

The impacts of low flow conditions on DOC concentrations were investigated at two temporal scales to explore the direct drought effects as well as the subsequent responses during the rewetting stage. The direct response captures changes in DOC, LMW DOC, C/N ratio, and SUVA$_{254}$ that occur during summer low flow periods. The post-drought effects investigated the change in these same response variables after the first rewetting. We similarly investigated drought effects on groundwater DOC, LMW DOC, C/N ratio, and SUVA$_{254}$ in forested riparian and mire soils during summer low flows and post-drought after the first rewetting.

The effects of prolonged summer low flow on both surface water and groundwater DOC were determined by first averaging the DOC concentrations during the summer period (Jul–August) in each catchment for each year. We then calculated the difference between individual summer averages and the long-term summer average for the 17 years (Supplementary Table 2) and expressed as percentage change, following:

$$DOC_d = \left( \frac{DOC_a - DOC_b}{DOC_b} \right) \times 100 \qquad (1)$$

where the drought effect (DOC$_d$) was determined as the percentage difference in DOC in each summer (DOC$_a$) compared with the long-term summer average (DOC$_b$). We used linear regression with the number of low flow days in the summer period to test the prediction that average DOC concentrations would decline with drying severity (Fig. 2a). We used a similar regression approach to test whether DOC in forest and mire groundwater were also affected by drought severity by comparing values measured in the summer to pre-drought values (the long-term summer average).

The effect of prolonged summer low flows on the initial flush of DOC (DOC$_d$) upon rewetting was estimated as the difference between DOC measured after the longest low flow period (when there was a rain event that caused an increase in runoff) (DOC$_a$) and DOC measured before the onset of low flows each summer (DOC$_b$) as in the Eq. (1). Here, we expected that DOC produced in soils during the dry periods would be flushed out in the first rewetting event, reflecting processes hindering DOC consumption or favoring DOC production during drought. We used linear regression with the number of low flows in each summer to test whether changes in DOC were related to the duration of the low flow periods. For groundwater analysis, similar calculations were used to show the differences between the post-summer (September) and pre-summer (end of May–June) values and to ask whether there were any post-drought effects on groundwater in either the forest soils or mires. All differences were expressed as percentage changes from the pre-drought concentrations. We used the linear regression with the number of flow days below the 0.1 mm per day threshold to test whether the magnitude of these (lagged) effects was related to the severity of summer drying.

We also used the specific discharge data to calculate normalized DOC concentration as a conservative measure to test whether discharge conditions at the time of sampling could bias results (Supplementary Fig. 1). However, both drought and post-drought analyses using normalized DOC concentrations were qualitatively the same as the results using unweighted DOC concentrations, indicating that changes in DOC observed could not be explained by the variation in discharge at the time of sampling. In addition, DOC concentration data

were combined with hydrological measurements to estimate seasonal and annual DOC fluxes for each site. We did this by multiplying specific daily discharge from individual catchments with linearly interpolated daily concentration data to generate fluxes at seasonal (summer and autumn) and annual time scales. Summer and annual fluxes were expressed as a percentage of the long-term average, while autumn fluxes were expressed as percentage change from pre-drought June averages. The linear regression with the number of low flow days in summer was then used to test whether changes in DOC exports at the different time scales were related to the duration of low flow periods.

### Drought and post-drought effects on LMW DOC, C/N ratio, and SUVA$_{254}$

The drought and post-drought effects analysis for LMW DOC, C/N ratio, and SUVA$_{254}$ followed the same procedure as used in the DOC modeling for stream water and groundwater data. With these three independent variables of carbon character, we expected that changes that occur as a result of the prolonged dry summer periods would be reflected in the quality of the carbon (LMW DOC, C/N ratio, and SUVA$_{254}$) when soils are flushed during rewetting. In these analyses, we expected that if the drought and post-drought effects on DOC are purely hydrological, there would be no change in the LMW DOC, C/N ratio, and SUVA$_{254}$ indicating that the quality is unaffected by prolonged dry periods. Here, an increasing trend in the LMW DOC signifies a shift to carbon with lower molecular weight and higher bacterial productivity[24,69] while increasing C/N ratio may indicate increasing biodegradability of DOC[70]. Higher SUVA$_{254}$ indicates plant litter or soil sources that are more aromatic[28]. Conversely, decreasing LMW DOC indicates a shift to more aromatic compounds with higher molecular weight, while a lower C/N ratio indicates DOC supply from more strongly processed soils at lower depths[26]. The lower SUVA$_{254}$ indicates microbial sources of carbon that are less aromatic.

### Slope relationships with land cover and catchment area

To better understand the drivers of drought response, we tested how the slope of the regression relating DOC change to drought severity at each site varied with catchment features. Here we used the number of low flow days as a measure of drought severity. The slope of this relationship represents the rate of change in DOC concentration as drought severity increases at each stream, thus providing an integrative assessment of drought sensitivity. We used stepwise multiple linear regression (in Minitab 18.1) to test whether this response varied among streams as a function of sub-catchment sizes, as well as the percentage of peat soils, forest, lake, and sedimentary soil cover in each sub-catchment.

### Data availability

The data for DOC quantity and quality as well as the discharge are available from the Unity Svartberget Data, https://franklin.vfp.slu.se/). The data sets used in this study have been deposited in Figshare Digital Repository https://figshare.com (https://doi.org/10.6084/m9.figshare.20176625).

### Code availability

The R code used to generate the figures in this manuscript has been deposited in Figshare Digital Repository https://figshare.com (https://doi.org/10.6084/m9.figshare.20176625).

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

## Acknowledgements

We thank the funders of the Krycklan Catchment Study, including the Swedish Science Foundation (VR), Swedish Infrastructure for Ecosystem Science (SITES), the VR X-stream project, Kempe Foundation, the Swedish Research Council for Sustainable Development (FORMAS), and the Swedish Nuclear and Waste Company (SKB).

## Author contributions

T.T. created the models and prepared the manuscript. R.S. was part of designing the study, interpreting the results, and writing the manuscript. H.L. initiated and conceptualized the study, provided the data, assisted in interpreting and writing the manuscript, and acquired the funding.

## Funding

## Competing interests

The authors declare no competing interests.
