## [Peer Review File · Nature Communications]

The emerging role of drought as a regulator of dissolved organic carbon in the boreal landscapesREVIEWER COMMENTS

Reviewer #1 (Remarks to the Author):

The authors highlight the role of seasonal drought for DOC mobilization in boreal landscapes. They show that in-stream DOC concentrations in 13 catchments are lower than the average during drought periods but higher than the average after the drought periods. Changes have also been observed in two chemical DOC characteristics. As it has already been shown in many other studies that DOC is largely controlled by the hydrological conditions, the overall statement is no surprise. However, the authors provide a large data set, both spatially and temporally, which nicely shows the broader relevance of the topic instead of only observing one catchment over a short time scale as done in several earlier studies. The manuscript is very well written and structured, and the results are presented in clear and nice figures. I think the manuscript is suitable for publication. However, the authors should carefully address some points.

General comments

- 1) It has been known for some time that in-stream DOC concentrations vary strongly with discharge (e.g., Lewis and Grant, 1979; Hornberger et al., 1994), and recently it has been shown that there are even diurnal variations (Tunaley et al., 2018). It is a weak point of the manuscript that the samples were taken without taking into account the current discharge. The authors compared the averaged summer DOC concentration to the number of low flow days during the respective summer. As far as I understand, the samples could have been taken before, during or after an event. The discharge conditions could be biasing the DOC concentrations. This problem could be solved by normalizing the DOC concentrations to discharge values (at the sampling time).
- 2) The authors chose the absorption ratio 254/365 and the CN ratio as parameters to investigate the chemical character of the DOC. I am not convinced that these parameters can really be seen as 'key characteristics', as the authors stated. In older as well as in recent studies, SUVA₂₅₄ was used to characterize DOC via absorbance spectrometry as it is a good proxy for aromaticity. Only a few studies have used 254/365 and C/N ratios to characterize DOC. Recently, fluorescence spectrometry gained in importance for the investigation of DOC characteristics. Is there a possibility of including other parameters for DOC characterization? If not, please clarify why you think that the 254/365 and C/N ratios are good parameters to characterize DOC in your study.
- 3) Also, absorbance was analyzed for only a very small fraction of samples. How can you ensure that these samples are representative?
- 4) I think it would be helpful to have a table in the supplementary material showing the averaged DOC concentrations as well as the values of 254/365 and C/N ratio. This would allow the reader to check the range of concentrations compared to other studies and to understand how relevant the observed changes are.

5) Several other papers investigated the link between DOC export and drought. However, they mostly study a specific catchment. As I have already argued, I think the advantage of the presented study is the large data set over a range of catchments and a large time scale. Some of the following studies might still be relevant for this study and helpful for the discussion: Blaurock et al. (2021), Clark et al. (2012), Toberman et al. (2008), Romaní et al. (2006)

Specific comments

L39, L151, L244 Some references are shown by name in the text instead of by a number. Maybe this is a formatting issue?

L43 Support your statement with a reference.

L83 – 84 This statement may be unnecessary. Is it important to highlight these two years?

L87 According to Figure 1, there are only two summers for which this is true. Therefore, this statement seems a bit over the top. Alternatively, write 'the two summers with more...'

L88 I do not understand the information in the brackets. What is 'small to larger catchments' and is all this information still referring to the two summers with more than 40 low flow days?

L89 I think there is a bracket missing.

L107 Maybe mention earlier what the CN ratio actually is useful for, e.g. in L69

L 123 – 124 I agree that DOC is accumulating during drought periods and therefore export increases after drought. However, I am not sure if the argument that DOC production increases during drought periods is generally true, as research has shown that mineralization and production rates are usually higher in wet soils (Borken and Matzner, 2009; Muhr et al., 2010). In the cited study (Ref 28), the authors argue that the soils at their study site are characterized by high soil moisture even during the dry situation, therefore a high DOC production can be maintained and lead to a high DOC export after drought. Is this true for the here investigated catchments as well?

L127 Krycklan is mentioned here for the first time. Either leave the name out of the text (except for the methods) or mention shortly that this is the name of your catchment area.

L153 – 154 It is not completely clear why more LWM DOC gets exported upon rewetting. Maybe insert a sentence with the information that organic soils are characterized by a higher LWM content and provide a reference.

L156 Space missing between ca. and 0.2

L165 The term 'time frame' does not sound right here. I would expect the next paragraph to talk about time scales, but it is rather related to spatial scales.

L171 – 179 Are you referring to longitudinal connectivity along the stream network? Do you mean by 'losses in connectivity' that streams are intermittent? Headwaters are also often characterized by DOC rich riparian zones, which might get disconnected during drought (lateral connectivity). I would expect that the disconnection and subsequent reconnection would lead to strong DOC responses in the

headwaters due to the proximity of important DOC sources. Further downstream, the signals would then arrive delayed and attenuated independently of the connection to deep groundwater sources.

L202 – 205 This sentence is a bit convoluted. Try to reformulate.

L229 Does this reference only refer to the large area or to the whole sentence? If the latter applies, please exclude from brackets

L236 Mention Scots pine first as it is more important than Norway spruce.

L239 I would remove 'the'. But why does spring only refer to mid-April? Is this just the time, when the contribution of snowmelt was measured? Then you could write 'In mid-April, snowmelt accounts...'?

L258 Please add the months to which you refer when writing about summer

L311 What about August?

L332 In which month does pre-summer start? Maybe clarify once at the beginning of the method section which months belong to which season. This would also help to clarify the two previous comments

L333 Supplementary Figure S2 instead of Fig. S2, to stay consistent

L347 'indicate' instead of 'indicates'

Figure 1a): Why are some years missing? How does this figure add value? Maybe Figure 1b would be sufficient?

Figure 3: Please check the terms used in the brackets to stay consistent. I suppose 'blue line' should be 'blue dots' and 'blue' should be 'blue line' (the same for orange).

Figure 5: Add the missing unit to L553

Figure 6: I suggest that you put a title on top of the figures, left 'drought' and right 'post-drought' as done in Figures 2 and 4. I would make the message much clearer.

References

Blaurock, K., Beudert, B., Gilfedder, B. S., Fleckenstein, J. H., Peiffer, S., and Hopp, L.: Low hydrological connectivity after summer drought inhibits DOC export in a forested headwater catchment, *Hydrol. Earth Syst. Sci.*, 25, 5133–5151, <https://doi.org/10.5194/hess-25-5133-2021>, 2021.

Borken, W. and Matzner, E.: Reappraisal of drying and wetting effects on C and N mineralization and fluxes in soils, *Global change biology*, 15, 808–824, <https://doi.org/10.1111/j.1365-2486.2008.01681.x>, 2009.

Clark, J. M., Heinemeyer, A., Martin, P., and Bottrell, S. H.: Processes controlling DOC in pore water during simulated drought cycles in six different UK peats, *Biogeochemistry*, 109, 253–270, <https://doi.org/10.1007/s10533-011-9624-9>, 2012.

Hornberger, G. M., Bencala, K. E., and McKnight, D. M.: Hydrological controls on dissolved organic carbon during snowmelt in the Snake River near Montezuma, Colorado, *Biogeochemistry*, 25, 147–165, <https://doi.org/10.1007/BF00024390>, 1994.

Lewis, W. M. and Grant, M. C.: Relationship between stream discharge and yield of dissolved substances from a Colorado Mountain watershed, *Soil Science*, 128, 353–363, 1979.

Muhr, J., Franke, J., and Borken, W.: Drying-rewetting events reduce C and N losses from a Norway spruce forest floor, *SOIL BIOLOGY & BIOCHEMISTRY*, 42, 1303–1312, <https://doi.org/10.1016/j.soilbio.2010.03.024>, 2010.

Romaní, A. M., Vázquez, E., and Butturini, A.: Microbial availability and size fractionation of dissolved organic carbon after drought in an intermittent stream: biogeochemical link across the stream-riparian interface, *Microbial ecology*, 52, 501–512, <https://doi.org/10.1007/s00248-006-9112-2>, 2006.

Toberman, H., Evans, C. D., Freeman, C., Fenner, N., White, M., Emmett, B. A., and Artz, R. R.: Summer drought effects upon soil and litter extracellular phenol oxidase activity and soluble carbon release in an upland Calluna heathland, *Soil Biology and Biochemistry*, 40, 1519–1532, <https://doi.org/10.1016/j.soilbio.2008.01.004>, 2008.

Tunaley, C., Tetzlaff, D., Wang, H., and Soulsby, C.: Spatio-temporal diel DOC cycles in a wet, low energy, northern catchment: Highlighting and questioning the sub-daily rhythms of catchment functioning, *Journal of Hydrology*, 563, 962–974, <https://doi.org/10.1016/j.jhydrol.2018.06.056>, 2018.

Reviewer #2 (Remarks to the Author):

Review to the manuscript “The emerging role of drought as a regulator of dissolved organic carbon in the boreal landscapes” by Tejshree Tiwari, Ryan A. Sponseller and Hjalmar Laudon

The manuscript of Tejshree Tiwari, Ryan A. Sponseller and Hjalmar Laudon is based on a 17-year record of DOC concentration and absorbance measurements in the Krycklan catchment in northern Sweden. This impressive data set enables profound statements on altered DOC pattern due to climate change and predicted enhanced drought periods. The manuscript highlights that drought periods lead to lower DOC concentrations predominantly due to the hydrologic disconnection of the top soils within the catchment. However, during the post-drought rewetting period DOC concentrations are much higher compared to a long-term average, due to DOC production during drought, and add up to a shift in the annual DOC pattern. While the general finding of lower DOC concentrations during summer drought periods and a DOC pulse after rewetting is not entirely new, it has never been shown on such an extensive data set. The study design and generated data set complies with scientific standards. Even though applied statistics are not sophisticated it generates a clear message. The manuscript is well written, and my points as raised below are rather minor ones. My main issues are 1) at some points more precise wording 2) more detailed description or emphasis of the importance and consequences

that this shift in DOC pattern has for the aquatic system 3) incorporation of other studies to DOC production or dry/wet catchment conditions. Please see the more detailed descriptions below.

One of my raised points is to be more detailed in the description of consequences of the changed DOC concentration pattern. You phrase “far-reaching” or “widespread” consequences in the abstract and introduction section, as well as in the Results and Discussion chapter. However, what this exactly means remains unclear. Being more specific at this point would also highlight the importance of this study.

The study would also benefit at several parts from the inclusion of more references to studies on DOC production under dry conditions (Line 37, Line 55 to 65) or short-term studies on DOC export and drought. Maybe there are no Scandinavian or boreal studies to this, but in general those studies are available, e.g. from UK, the Experimental Lake Area (ELA), or other parts of Europe.

One thought to your Network scale responses chapter: Checking the method section, I want to add a comment to your sampling resolution. Having a long-term monitoring over decades is very rare and I totally agree that this is a very valuable data set, which has been generated with great efforts. I just wonder if you might give a comment in your discussion concerning weaknesses due to the sampling resolution. A weekly sampling resolution might bias the rewetting signal due to event dynamics in DOC quantity and quality. It is very dependent at which time of the rewetting event the sample is taken. Maybe this is ok on a long-term perspective? Can you give a comment to that concern?

This might also play a role at the small versus large catchment response: Can this also be biased by sampling resolution? Large catchments might generate a slower, longer response than small ones where you might miss shorter response signals by sampling on a weekly resolution?

Furthermore, I would recommend shortening the “Conceptualizing” section as there are many repetitions from the previous sections.

An interesting addition to the presented evaluation could be whether the annual DOC export (budget) from the catchment changes due to enhanced drought periods. Is it possible for you to state whether dry years in total export more or less DOC compared to a long-term average? It would be also interesting if there is a lagged effect in the following post-drought year. This would also enable you to make a statement whether there can be expected a greater soil carbon/DOC loss in future due to enhanced drought periods.

Specific comments:

Line 29: “potentially far-reaching consequences” Can you be more specific here?

Line 37: maybe some more references? Studies from ELA, UK, Central or Eastern Europe?

Line 39 and others: check reference embedding

Line 39 ff: “This knowledge gap...lead to dramatic and unexpected impacts...” From my point of view a knowledge gap might cause unexpected impacts, but dramatic? Please check the wording

Line 47: I would delete “northern”. Even though your catchment is located in northern landscapes this statement is generally valid

Line 69: I think the writing “carbon to nitrogen ratio (C/N ratio)” is more commonly used

Line 73: please introduce relevance of LMW DOC, and C/N ratio. Why do you expect this? (It gets clear later in the text, but it is a riddle at this point)

Line 100: “...increase DOC supply...” by higher production? Maybe briefly add the possible causes

Line 103: You sometimes refer to “drought episodes”, sometimes “dry periods”. Please be consistent.

Line 108, 109: a bit confusing. Whether N species are reduced or not is non-relevant to the C/N ratio. Oxidized N species at the top soil could also contribute to a lower C/N. Maybe you should make your point clearer here (higher levels of inorganic N?).

Line 119 ff: What consequences do you expect exactly? Please delineate this with more details. E.g. longer residence times and higher N availability or higher DOC reactivity? Anoxia? For this you might also refer to ELA studies or more recent studies on increased residence times and anoxia in surface water due to lower flow and higher temperatures.

Line 134: LMW DOC increased linearly compared to what?

Line 140: “expressed to the severity” expressed as...? Can you help the reader by a short complement? From your figures and context, I guess drought severity is measured as number of low flow days, but you state this nowhere (e.g. “Drought severity” used in L66, 86, 89, 104). At some point you also use the phrase “drought magnitude”, “drought duration”. Is this all the same?

Reviewer #3 (Remarks to the Author):

This is an interesting and well written paper on the increasing role of drought in regulating DOC concentrations in boreal rivers. I enjoyed reading the paper and the authors have collected a useful dataset. I am less certain on the impact of this paper, and hence its publication in a nature journal, as the authors are testing known results in a new location. I would recommend the paper for publication as is in a lower ranking journal but feel the authors may need to do more to justify Nature Comms.

The hypothesis (L70 onwards) could be restated as ‘we expect boreal rivers to be the same as non-boreal rivers’ which is what was found. I understand the incidence of drought in these landscapes is somewhat novel, and expected to increase, and therefore suggest the authors perform an analysis of how projected climate change could increase DOC flux from these catchments. Essentially, combine their excellent Fig 7 with a projection of drought frequency/severity in the future so we can understand the scale of the issue.

Furthermore, the authors have an extensive DOC concentration dataset as well as flow data. Why no calculation of variation in DOC load or flux between catchments, only concentration?

Could differences in residence time in the catchment explain some of the variation between small and large catchments found?

L166 onwards is very interesting as it seems to negate the 'river as chemostat' hypothesis.

L191 suggests peat catchments are more resilient to droughts (higher water storage capacity and lower evaporative losses), and therefore the catchments analysed did not experience the same drought severity in terms of e.g. WT depth change. Does this not affect the analysis across catchments if we're effectively seeing a milder drought in some?

Inorganic N is <9% of total N and is discarded due to limited data at some sites. Could you perhaps justify this further by showing for the sites there are data that this does not vary with drought or season to give use more confidence this does not explain some of the TN effect?

Overall, this is a good paper but I feel to be high impact the authors may have to do a bit more as the results are largely expected.

REVIEWER COMMENTS

Response to reviewer #1 comments

The authors highlight the role of seasonal drought for DOC mobilization in boreal landscapes. They show that in-stream DOC concentrations in 13 catchments are lower than the average during drought periods but higher than the average after the drought periods. Changes have also been observed in two chemical DOC characteristics. As it has already been shown in many other studies that DOC is largely controlled by the hydrological conditions, the overall statement is no surprise. However, the authors provide a large data set, both spatially and temporally, which nicely shows the broader relevance of the topic instead of only observing one catchment over a short time scale as done in several earlier studies. The manuscript is very well written and structured, and the results are presented in clear and nice figures. I think the manuscript is suitable for publication. However, the authors should carefully address some points.

Response: We are grateful to the reviewer for the comments and detailed reference list, which has widened the scope of our research in terms of dry/rewetting effects on DOC. On the point of DOC being largely controlled by hydrological conditions, while we agree that many studies have reported positive relationships between discharge and DOC [Clark *et al.*, 2012; Pastor *et al.*, 2003; Zarnetske *et al.*, 2018], most of these have focused on changes in concentrations from baseflow to floods (i.e., ‘flood responses’). However, in our study, the focus was in the opposite direction, that is: what happens to DOC concentration from baseflow to drought conditions? In this context, we know much less about the hydrological relationship. Indeed, despite the frequent assumption that DOC concentrations increase with discharge, evidence suggests that this is not typically linear across the full range of flow conditions. For example, in a survey of nearly 300 streams/rivers in France, [Moatar *et al.*, 2017] found that DOC concentrations continued to decline as discharge dropped from median to (ostensibly) drought levels. For most sites, evident hydrological control over DOC concentrations was only observed from median to high discharge. Thus, while perhaps not completely surprising, the persistent declines in concentration as drought severity increases reported in our paper are in fact not universally observed and point to particularly strong hydrological influences over the full seasonal supply of DOC streams.

However, the comments and suggestions on the importance of discharge also encouraged us to strengthen our analysis by calculating export changes during different seasons, which are now included both as a short text and in the supplementary information (Line 91-96, 98-100, 116-120, 399-412 Supplementary Figure S1). Additionally, we have also cross-checked the results with normalized concentrations to discharge as suggested. Finally, the suggestion of using other more commonly used DOC quality variables has prompted us to include SUVA₂₅₄ in the analysis which adds both rigor and a landscape dimension to drought impact on biogeochemistry in boreal streams in support of our previous results. Overall, we think that these changes have strengthened the manuscript.

General comments

- 1) It has been known for some time that in-stream DOC concentrations vary strongly with discharge (e.g., Lewis and Grant, 1979; Hornberger et al., 1994), and recently it has been shown that there are even diurnal variations (Tunaley et al., 2018). It is a weak point of the manuscript that the samples were taken without taking into account the current discharge.

Response: We agree that many studies have explored discharge-DOC concentration relationships, yet the vast majority of these have emphasized high flow rather than low flow responses (see response above). While some of our results (e.g., patterns in bulk DOC) suggest that chemical responses to flood and drought may be different sides of the same coin in this landscape, it does not necessarily have to work this way, and in fact, other responses variables (related to DOC character) revealed changes during dry periods that are observed during the re-wetting phase. We also agree on the importance of accounting for discharge at the point of sampling DOC following summer drought events and have now made steps towards including runoff variation (Supplementary Figure S1). We address this specifically in response to comment 2 (below) where we used normalized DOC concentrations to discharge to recreate this study and in response to Reviewer 2 (comment 6) where we in detail investigate variation in export. Using normalized discharge, we found magnified patterns that are consistent with our previous results using concentrations only (Figure R1). In relation to seasonal exports, the analysis is consistent with the shifts in the amount of DOC showing lower quantities during the summer followed by high amounts in the autumn (Figure R6). Following the suggestions by the three reviewers, we have included these analyses in the manuscript which has strengthened our conceptualization of drought impacts on the stream chemistry section (Lines 100-103, 124-129, 417-430). We believe that these additions increase the robustness of the drought impact analysis on DOC seasonal distribution as discharge has been better integrated into the manuscript.

- 2) The authors compared the averaged summer DOC concentration to the number of low flow days during the respective summer. As far as I understand, the samples could have been taken before, during, or after an event. The discharge conditions could be biasing the DOC concentrations. This problem could be solved by normalizing the DOC concentrations to discharge values (at the sampling time).

Response: Thank you for the suggestion! We have tested this by including specific discharge from each catchment to normalize DOC concentrations (Figure R1 A, B). Both drought and post-drought analysis were recalculated with the new DOC data normalized to discharge as suggested. In Figure R1 (A, B), the results are consistent with our previous results based on DOC concentrations only (Figure R1C, D), with the difference that the drought effect became larger using the normalized dataset. However, to be conservative, we continue to use the unweighted dataset in the manuscript as it shows a similar result without the additional uncertainties of normalizing the discharge. However, we have added this point to the manuscript which we hope addresses any issues of discharge bias (Lines 91-96, 98-100, 116-120, 399-412).

On the point that the samples could be taken, before, during, or after an event, we do not expect that this should significantly affect the outcome of our results. In the drought analysis, we used an average of the drought period for each year that is consistent with regular sampling every two weeks reflecting a variety of flow conditions. By using the range of summer conditions, we believe that we have a large enough confidence interval to represent the flow conditions across the years without biasing the timing of sampling.

Figure R1 Drought and Post-drought effects using calculated Normalised DOC data to discharge (A,B respectively where drought r^2 range 0.31-0.68, $p < 0.05$ and post-drought r^2 range 0.20-0.57, $p < 0.05$) and DOC concentrations (C and D respectively where drought r^2 range 0.29-0.65, $p < 0.05$ and post-drought r^2 range 0.28-0.67, $p < 0.05$).

- 3) The authors chose the absorption ratio 254/365 and the C/N ratio as parameters to investigate the chemical character of the DOC. I am not convinced that these parameters can really be seen as ‘key characteristics’, as the authors stated. In older as well as in recent studies, SUVA₂₅₄ was used to characterize DOC via absorbance spectrometry as it is a good proxy for aromaticity. Only a few studies have used 254/365 and C/N ratios to characterize DOC. Recently, fluorescence spectrometry gained in importance for the investigation of DOC characteristics. Is there a possibility of including other parameters for DOC characterization? If not, please clarify why you think that the 254/365 and C/N ratios are good parameters to characterize DOC in your study.

Response: Following the reviewer's suggestion of including other parameters, we have investigated changes in SUVA₂₅₄ as yet another test of the quality of DOC in the manuscript (Figure R2). SUVA₂₅₄ is also a relatively common test of the compositions of DOC in streams where higher values indicate higher contents of aromatic carbon that is generally lower in BGE compared to the aliphatic compounds [Weishaar *et al.*, 2003]. The drought analysis showed a distinction in the forest and mire-dominated catchments where higher SUVA was seen in the mire catchment compared to the forest catchment during drought (Figure R2 c). This is in line with our LMW DOC (Abs 245/365) results and previous research in this catchment [Agren *et al.*, 2008] which showed high SUVA₂₅₄ in wetland-dominated streams and low SUVA₂₅₄ in forest-dominated streams suggesting a change in the composition of carbon from higher molecular weight to lower molecular weight and from aromatic to more aliphatic compounds during low flows. This decreasing SUVA₂₅₄ with longer durations of low flow suggests a decrease in aromatic carbon in the forested catchments as organic-rich layers are disconnected which is also consistent with the LMW DOC and C/N ratio results. We have now included this in the results and discussion.

More generally, we are aware of the limitations of these metrics as proxies for DOC quality. Yet, more refined/detailed measures do not exist anywhere for such a long time series of data for so many sites. The reason for using LMW DOC and C/N ratio is that these have been used as proxies for DOC quality, including in this same study system. Abs ratio A_{254}/A_{365} is a simple technique that is negatively correlated with dissolved humic substances [Dahlen *et al.*, 1996]. C/N ratio has been a traditional way of expressing substrate quality in both terrestrial [Kalbitz *et al.*, 2000] and aquatic environments where a low C/N ratio reflects recalcitrance to degradation at greater soil depth and is a regulator of bacterial growth efficiency in aquatic systems [Kroer, 1993]. Both LMW DOC and C/N ratios have been used to show the importance of carbon derived from various landscape types in bacterial biomass growth [Berggren *et al.*, 2007; 2010]. We have added this description to the introduction and hope that this clarifies the importance of using these variables to detect quality changes in DOC (Line 67:77).

Figure R2 Changes in carbon character as measured by LMW DOC (Abs ratio A_{254}/A_{365})(a,b), C/N ratio (c,d) and SUVA₂₅₄ (e,f)

- 4) Also, absorbance was analyzed for only a very small fraction of samples. How can you ensure that these samples are representative?

Response: Concerning this reviewer's comment, we have consulted the Krycklan database for additional absorbance data. Due to instrument errors in the earlier years, much of the data could not be quality assured. However, we were able to expand the analysis by two additional years 2012 and 2013. With the larger data set, we see a clearer pattern in LMW DOC than previously, which is also consistent with the responses seen in the C/N ratio and DOC which have complete datasets.

- 5) I think it would be helpful to have a table in the supplementary material showing the averaged DOC concentrations as well as the values of 254/365 and C/N ratio. This would allow the reader to check the range of concentrations compared to other studies and to understand how relevant the observed changes are.

Response: Thank you for the comment, the table has now been included in supplementary Table S2 (Line 375).

- 6) Several other papers investigated the link between DOC export and drought. However, they mostly study a specific catchment. As I have already argued, I think the advantage of the presented study is the large data set over a range of catchments and a large time scale. Some of the following studies might still be relevant for this study and helpful for the discussion: Blaurock et al. (2021), Clark et al. (2012), Toberman et al. (2008), Romaní et al. (2006)

Response: We thank this reviewer for these references that support many of the concepts and ideas presented in our research. These references have now been embedded into our manuscript to support the different mechanisms or regulations of DOC specifically microbial processing and inhibition of phenol oxidase (Line 62, 64, 197, 207) during drought.

Specific comments

- 7) L39, L151, L244 Some references are shown by name in the text instead of by a number. Maybe this is a formatting issue?

Response: The references have been fixed (lines 41, 198, 279)

- 8) L43 Support your statement with a reference.

Response: The reference has been added (Line 46)

- 9) L83 – 84 This statement may be unnecessary. Is it important to highlight these two years?

Response: With this statement, we wanted to highlight the extremity of these dry years in relation to the previous dry years. The number of low flow days in these years was more than two times the standard deviation from the mean suggesting the extremity of these years according to the extreme value theory [*Ghil et al.*, 2011]

- 10) L87 According to Figure 1, there are only two summers for which this is true. Therefore, this statement seems a bit over the top. Alternatively, write ‘the two summers with more...’

Response: This suggestion has been made (Line 94)

11) L88 I do not understand the information in the brackets. What is ‘small to larger catchments’ and is all this information still referring to the two summers with more than 40 low flow days?

Response: We apologize for the unclear structure of the information in the brackets and now have made some changes to show that during the two years with the highest low days (>40 days) DOC concentration was 20-55% lower in all catchments. (Line 96)

12) L89 I think there is a bracket missing.

Response: This sentence has been restructured (Line 96)

13) L107 Maybe mention earlier what the CN ratio actually is useful for, e.g. in L69

Response: Thank you for the suggestion of clarifying the usefulness of the C/N ratio. We have also followed this suggestion and included explanations for using abs ratio and SUVA in defining DOC character in the introduction (Lines 73-75).

14) L 123 – 124 I agree that DOC is accumulating during drought periods and therefore export increases after drought. However, I am not sure if the argument that DOC production increases during drought periods are generally true, as research has shown that mineralization and production rates are usually higher in wet soils (Borken and Matzner, 2009; Muhr et al., 2010). In the cited study (Ref 28), the authors argue that the soils at their study site are characterized by high soil moisture even during the dry situation, therefore a high DOC production can be maintained and lead to a high DOC export after drought. Is this true for the here investigated catchments as well?

Response: We understand the concerns of the sources of DOC upon rewetting after drought might be related to the accumulation of previously produced DOC that is immobilized due to the change in the flow paths. However, our three independent quality analyses all suggested that changes to the DOC pool had occurred, indicating the enhanced concentration could also related to higher production during the dry periods.

However, we do agree that the intensity of the drought as measured by the number of low flow days in our research could not explain all the variability in DOC observed across the years and catchment. Additionally, the annual flux calculation of DOC did not show a significant trend in the increase in DOC however, seasonal changes in flux suggest respective changes with low DOC concentrations during summer and higher during autumn following prolonged droughts (Figure R4). As such, we agree that the size of the rewetting event may have some influence on the pulses observed in DOC as has been shown in previous works in the same catchments [Tiwari *et al.*, 2018; 2019] where higher spring exports could be explained by the size of the precipitation events in the previous summers and autumns which in inline with [Werner *et al.*, 2019]. On the same note, years with a small number of low flow days did indeed show higher DOC during the drought compared to the drier years (Figure 2a). However, these were not the years with the highest DOC upon rewetting suggesting possible exhaustion of DOC sources in these wetter years even if production and mineralization were indeed higher (Figure R4 Autumn).

On the point that mineralization and production of DOC were higher in wet soils, as presented by *Muhr et al.* [2010] and *Borken and Matzner* [2009], we found to be controversial as both studies used results from soil core experiments that have been subjected to drying and rewetting regimes in a laboratory to assess changes in CO₂. Both authors have indicated that these experiments cannot be extrapolated to ecosystem levels as they do not account for inhomogeneous changes in soil moisture, air temperature, and soil aggregation. In *Muhr et al.* [2010], the authors also suggested that the experiment may have been too short-term to see the mineralization pulse after rewetting.

In response to this question and previous comments (comments 1,2), we have included the export analysis in the supplementary Figure to support the concepts of changes in DOC exports as a result of prolonged summer droughts (Supplementary Figure S1, Lines 91-96, 98-100, 116-120, 399-412). We hope that with these additions, the influence of droughts on stream DOC becomes apparent.

- 15) L127 Krycklan is mentioned here for the first time. Either leave the name out of the text (except for the methods) or mention shortly that this is the name of your catchment area.

Response: Krycklan was replaced by “stream” (Line 168).

- 16) L153 – 154 It is not completely clear why more LWM DOC gets exported upon rewetting. Maybe insert a sentence with the information that organic soils are characterized by a higher LWM content and provide a reference.

Response: Thank you for the suggestion, we have added a sentence to this section “Collectively, these changes in DOC character can directly influence aquatic ecosystems by providing higher-quality organic substrates for heterotrophic bacteria [*van Hees et al.*, 2005]. (Line 203)

- 17) L156 Space missing between ca. and 0.2

Response: The typo was corrected (line 204)

- 18) L165 The term ‘time frame’ does not sound right here. I would expect the next paragraph to talk about time scales, but it is rather related to spatial scales.

Response: We apologize for the vague wording and now have rephrased this section (Line 217-220).

- 19) L171 – 179 Are you referring to longitudinal connectivity along the stream network? Do you mean by ‘losses in connectivity’ that streams are intermittent? Headwaters are also often characterized by DOC rich riparian zones, which might get disconnected during drought (lateral connectivity). I would expect that the disconnection and subsequent reconnection would lead to strong DOC responses in the headwaters due to the proximity of important DOC sources. Further downstream, the signals would then arrive delayed and attenuated independently of the connection to deep groundwater sources.

Response: Based on this reviewer’s comment, we have included the terms lateral and longitudinal connectivity to clarify the type of connectivity referred to (Line 228). In response to the second part of this comment, we would like to clarify that the changes we investigate show the relative changes of individual catchment DOC to long-term averages

and pre-drought concentrations, rather than absolute changes in concentration of DOC in headwaters to downstream which does indeed shows higher DOC in headwaters compared to downstream. We apologize for the misunderstanding and have now added a section to the manuscript to improve clarity.

20) L202 – 205 This sentence is a bit convoluted. Try to reformulate.

Response: We appreciate the comment and have restructured the sentence which now reads “Observed seasonal variation in amplitude of DOC, LMW DOC, C/N ratio and SUVA₂₅₄ suggests that while drought effects on stream biogeochemistry are direct and immediately (Figure 5a), the indirect, lagged effects extends beyond the duration of the direct disturbance (Figure 5b), with potentially different consequence down the aquatic network.” (Line 257-260)

21) L229 Does this reference only refer to the large area or the whole sentence? If the latter applies, please exclude it from brackets

Response: Thank you, the correction has been made, and the reference has been excluded from the brackets (Line 279)

22) L236 Mention Scots pine first as it is more important than Norway spruce.

Response: The correction has been made (line 286)

23) L239 I would remove ‘the’. But why does spring only refer to mid-April? Is this just the time, when the contribution of snowmelt was measured? Then you could write ‘In mid-April, snowmelt accounts...’?

Response: The correction has been made which now reads “In mid-April snowmelt accounts for approximately 40% of the annual runoff” (Line 289)

24) L258 Please add the months to which you refer when writing about summer

Response: The correction has been made (309)

25) L311 What about August?

Response: This was a typo, and was replaced by (-). This now reads “July-Aug” (Line 318)

26) L332 In which month does pre-summer start? Maybe clarify once at the beginning of the method section which months belong to which season. This would also help to clarify the two previous comments

Response: Agreed and we have made the addition as suggested: “For groundwater analysis, similar calculations were used to show the differences between the post-summer (Sept) and pre-summer (end of May-June) values and to ask whether there were any post-drought effects on groundwater in either the forest soils or mires”. (Line 322-324)

27) L333 Supplementary Figure S2 instead of Fig. S2, to stay consistent

Response: This figure has been omitted

28) L347 ‘indicate’ instead of ‘indicates’

Response: This section has been reworded “Conversely, decreasing LMW DOC indicates a shift to more aromatic compounds with higher molecular weight...” (Line 423)

Figures

- 1) Figure 1a): Why are some years missing? How does this figure add value? Maybe Figure 1b would be sufficient?

Response: Some years had no days where discharge was lower than 0.1 mm. We have removed Figure 1a as suggested.

- 2) Figure 3: Please check the terms used in the brackets to stay consistent. I suppose ‘blue line’ should be ‘blue dots’ and ‘blue’ should be ‘blue line’ (the same for orange).

Response: The correction has been made and the colours have been updated to reflect dry conditions as red and wet conditions as blue. Thank you for your comment.

- 3) Figure 5: Add the missing unit to L553

Response: The correction has been made to the figure legend which is now supplementary Figure S2 (correction 2-2.25 m).

- 4) Figure 6: I suggest that you put a title on top of the figures, left ‘drought’ and right ‘post-drought’ as done in Figures 2 and 4. I would make the message much clearer.

Response: The correction has been made to the figure which is now Supplementary Figure S3.

References

- 1) Blaurock, K., Beudert, B., Gilfedder, B. S., Fleckenstein, J. H., Peiffer, S., and Hopp, L.: Low hydrological connectivity after summer drought inhibits DOC export in a forested headwater catchment, *Hydrol. Earth Syst. Sci.*, 25, 5133–5151, <https://doi.org/10.5194/hess-25-5133-2021>, 2021.
- 2) Borken, W. and Matzner, E.: Reappraisal of drying and wetting effects on C and N mineralization and fluxes in soils, *Global change biology*, 15, 808–824, <https://doi.org/10.1111/j.1365-2486.2008.01681.x>, 2009.
- 3) Clark, J. M., Heinemeyer, A., Martin, P., and Bottrell, S. H.: Processes controlling DOC in pore water during simulated drought cycles in six different UK peats, *Biogeochemistry*, 109, 253–270, <https://doi.org/10.1007/s10533-011-9624-9>, 2012.
- 4) Hornberger, G. M., Bencala, K. E., and McKnight, D. M.: Hydrological controls on dissolved organic carbon during snowmelt in the Snake River near Montezuma, Colorado, *Biogeochemistry*, 25, 147–165, <https://doi.org/10.1007/BF00024390>, 1994.
- 5) Lewis, W. M. and Grant, M. C.: Relationship between stream discharge and yield of dissolved substances from a Colorado Mountain watershed, *Soil Science*, 128, 353–363, 1979.
- 6) Muhr, J., Franke, J., and Borken, W.: Drying-rewetting events reduce C and N losses from a Norway spruce forest floor, *SOIL BIOLOGY & BIOCHEMISTRY*, 42, 1303–1312, <https://doi.org/10.1016/j.soilbio.2010.03.024>, 2010.
- 7) Romaní, A. M., Vázquez, E., and Butturini, A.: Microbial availability and size fractionation of dissolved organic carbon after drought in an intermittent stream: biogeochemical link across the stream-riparian interface, *Microbial ecology*, 52, 501–512, <https://doi.org/10.1007/s00248-006-9112-2>, 2006.
- 8) Toberman, H., Evans, C. D., Freeman, C., Fenner, N., White, M., Emmett, B. A., and Artz, R. R.: Summer drought effects upon soil and litter extracellular phenol oxidase activity and soluble carbon release in an upland *Calluna* heathland, *Soil Biology and Biochemistry*, 40, 1519–1532, <https://doi.org/10.1016/j.soilbio.2008.01.004>, 2008.
- 9) Tunaley, C., Tetzlaff, D., Wang, H., and Soulsby, C.: Spatio-temporal diel DOC cycles in a wet, low energy, northern catchment: Highlighting and questioning the sub-daily rhythms of catchment functioning, *Journal of Hydrology*, 563, 962–974, <https://doi.org/10.1016/j.jhydrol.2018.06.056>, 2018.

Response to reviewer #2 comments

Review to the manuscript “The emerging role of drought as a regulator of dissolved organic carbon in the boreal landscapes” by Tejshree Tiwari, Ryan A. Sponseller, and Hjalmar Laudon

The manuscript of Tejshree Tiwari, Ryan A. Sponseller, and Hjalmar Laudon is based on a 17-year record of DOC concentration and absorbance measurements in the Krycklan catchment in northern Sweden. This impressive data set enables profound statements on altered DOC patterns due to climate change and predicted enhanced drought periods. The manuscript highlights that drought periods lead to lower DOC concentrations predominantly due to the hydrologic disconnection of the topsoils within the catchment. However, during the post-drought rewetting period DOC concentrations are much higher compared to a long-term average, due to DOC production during drought, and add up to a shift in the annual DOC pattern. While the general finding of lower DOC concentrations during summer drought periods and a DOC pulse after rewetting is not entirely new, it has never been shown on such an extensive data set. The study design and generated data set comply with scientific standards. Even though applied statistics are not sophisticated it generates a clear message. The manuscript is well written, and my points as raised below are rather minor ones. My main issues are 1) at some points more precise wording 2) more detailed description or emphasis of the importance and consequences that this shift in DOC pattern has for the aquatic system 3) incorporation of other studies to DOC production or dry/wet catchment conditions. Please see the more detailed descriptions below.

Response: The authors thank the reviewer for the positive feedback on the manuscript and for acknowledging the simplicity of the analysis. We have improved the wording further and have been more precise in the writing style, and added a section on the importance and consequences of the seasonal shifts in DOC patterns. We have also widened the reference on drying and rewetting conditions (see below for more detailed responses).

General Comments

1. One of my raised points is to be more detailed in the description of consequences of the changed DOC concentration pattern. Your phrase “far-reaching” or “widespread” consequences in the abstract and introduction section, as well as in the Results and Discussion chapter. However, what this exactly means remains unclear. Being more specific at this point would also highlight the importance of this study.

Response: Thank you for the comment, we have addressed this in the manuscript and have been more detailed in the terminology and discussion points (Line 29, 53-55).

2. The study would also benefit at several parts from the inclusion of more references to studies on DOC production under dry conditions (Line 37, Line 55 to 65) or short-term studies on DOC export and drought. Maybe there are no Scandinavian or boreal studies to this, but in general those studies are available, e.g. from UK, the Experimental Lake Area (ELA), or other parts of Europe.

Response: We appreciate the comment and have now added supporting literature in similar biomes including “[*Borken and Matzner*, 2009; *Creed et al.*, 2018; *Fenner and Freeman*, 2011; *Toberman et al.*, 2008; *Werner et al.*, 2019]” Thank

you for the suggestion, we hope that the additional references add further support to our study. Line 45-77

3. One thought to your Network scale responses chapter: Checking the method section, I want to add a comment to your sampling resolution. Having a long-term monitoring over decades is very rare and I totally agree that this is a very valuable data set, which has been generated with great efforts. I just wonder if you might give a comment in your discussion concerning weaknesses due to the sampling resolution. A weekly sampling resolution might bias the rewetting signal due to event dynamics in DOC quantity and quality. It is very dependent at which time of the rewetting event the sample is taken. Maybe this is ok on a long-term perspective? Can you give a comment to that concern?

Response: We appreciate this comment with regards to improving the robustness of the analysis despite the challenges of being unable to have a higher resolution dataset, which may better reflect the signal of individual events.

Definitely with such long-term datasets as ours, comparison among events over a longer period that varies in magnitude is possible with a standardized sampling regime across the 17 years. In regards to the network scale section, as we compare within catchment responses across the 17 years to variation among the 13 catchments, the consistent and regular sampling across the years and catchments provides a technique of comparing responses without the biases due to variations in flow or events.

Additionally, we have tested the sensitivity of our results to discharge by recreating the study of drought and post-drought using normalized concentrations. The results were consistent with the unweighted DOC concentration as addressed in reviewer 1, question 2 (Figure R1). We have made a note of this in the manuscript “This inter-annual variability in hydrology had clear effects on summer DOC concentrations, which declined as drought severity increased. The relationship between drought and DOC held whether or not concentrations were corrected for discharge on each sampling day”(Line 91-96, 98-100, 116-120, 399-412).

4. This might also play a role in the small versus large catchment response: Can this also be biased by sampling resolution? Large catchments might generate a slower, longer response than small ones where you might miss shorter response signals by sampling on a weekly resolution?

Response: Agreed, we have also addressed this topic for reviewer 1, question 2. We definitely cannot deny the effect that a finer grid sampling routine could improve the understanding of specific drought signals from individual catchments. However, maintaining a long-term dataset is both financially and time intensive which makes sampling on different dates not practical. However, we believe that by comparing catchment responses across such a long time period allowed us to identify the general trends in the dataset.

5. Furthermore, I would recommend shortening the “Conceptualizing” section as there are many repetitions from the previous sections.

Response: We have restructured this section entirely (Line 254).

6. An interesting addition to the presented evaluation could be whether the annual DOC export (budget) from the catchment changes due to enhanced drought periods. Is it possible for you to state whether dry years in total export more or less DOC compared to a long-term average? It would be also interesting if there is a lagged effect in the following post-drought year. This would also enable you to make a statement whether there can be expected a greater soil carbon/DOC loss in the future due to enhanced drought periods.

Response: Based on the flux analysis for the individual catchments, we cannot statistically detect any increase in the annual flux of carbon related to the number of low flow days in the summer. However, we observed an increase during the autumn which corresponds to the seasonal shift we noted in our manuscript. This has now been added to the supplementary Figure S1 (91-96, 98-100, 116-120, 399-412). However, we continue to use concentration in the manuscript as changes in flux, which are largely driven by hydrology and hence may not reflect the changes in the processes that we highlight using DOC concentrations and quality. For instance, the large quantity of water in the catchment caused by events masks the effects of concentrations in the catchments as discharge has an overarching role in regulating DOC [*Buffam et al.*, 2007; *Pacific et al.*, 2010]). As such, years like 2008 and 2012 with the highest precipitation and the summer with the lowest number of low flow days automatically export higher DOC on an annual basis. Looking at the fluxes seasonally where July and Aug. represent summer showed lower DOC fluxes in relation to longer durations of low flow while in the autumn we see a slight increase in the fluxes across the catchments. This supports the seasonal redistribution of DOC as we presented in the manuscript. However, on annual and inter-annual time scales, we can not statistically identify flux changes (Figure R3).

Figure R3 DOC exports calculated for the Summer, Autumn, and Annually

Specific comments:

7. Line 29: “potentially far-reaching consequences” Can you be more specific here?

Response: We have rephrased it with “Collectively, our results indicate an emerging shift in the seasonal distribution of DOC concentrations and character, which together operate as primary controls over the ecological and biogeochemical functioning of northern aquatic ecosystems” Lines 27-30

8. Line 37: maybe some more references? Studies from ELA, UK, Central or Eastern Europe?

Response: Thank you for the suggestions, the references have been added “[Acuna *et al.*, 2005; Borken and Matzner, 2009; Granados *et al.*, 2020]. (Line 40)

9. Line 39 and others: check reference embedding

Response: This error has been fixed (Line 41)

10. Line 39 ff: “This knowledge gap...lead to dramatic and unexpected impacts...” From my point of view a knowledge gap might cause unexpected impacts, but dramatic? Please check the wording

Response: This sentence was rephrased to “Yet, given the vast pools of organic matter that can be hydrologically mobilized in high latitude soils [Bradshaw and Warkentin, 2015], potential increases in drought frequency and intensified drying-rewetting cycles are likely to have pronounced effects on streams and rivers draining boreal and Arctic landscapes.” (Line 42-44)

11. Line 47: I would delete “northern”. Even though your catchment is located in northern landscapes this statement is generally valid

Response: Agreed, northern has been omitted. (Line 49).

12. Line 69: I think the writing “carbon to nitrogen ratio (C/N ratio)” is more commonly used

Response: We have now incorporated this into the manuscript

13. Line 73: please introduce the relevance of LMW DOC, and C/N ratio. Why do you expect this? (It gets clear later in the text, but it is a riddle at this point)

Response: We have used your suggestion and added these details to the introduction (Line 67-77).

14. Line 100: “...increase DOC supply...” by higher production? Maybe briefly add the possible causes

Response: This sentence has been restructured (Line 120:125)

15. Line 103: You sometimes refer to “drought episodes”, sometimes “dry periods”. Please be consistent.

Response: Drought episodes have been replaced with dry periods

16. Line 108, 109: a bit confusing. Whether N species are reduced or not is non-relevant to the C/N ratio. Oxidized N species at the topsoil could also contribute to a lower C/N. Maybe you should make your point clearer here (higher levels of inorganic N?).

Response: We apologize for the confusing sentence and have restructured “This response most likely reflects well-documented declines in soil C/N with depth in northern forests [*Marty et al., 2017*] and mires [*Kuhry and Vitt, 1996*], which results from long-term organic matter processing and potentially the accumulation of microbial N [*Rumpel and Kogel-Knabner, 2011*]. Thus, as the water table drops during extreme low flow periods, deeper soil strata characterized by more highly processed, lower C/N organic matter emerge as the sole source of dissolved organic matter to streams”. We hope that makes the point clearer. (Line 132:136)

17. Line 119 ff: What consequences do you expect exactly? Please delineate this with more details. E.g. longer residence times and higher N availability or higher DOC reactivity? Anoxia? For this, you might also refer to ELA studies or more recent studies on increased residence times and anoxia in surface water due to lower flow and higher temperatures.

Response: Thank you for the comment, we have now written more details in this section with references to ELA studies as well as more recent studies. This now reads “For headwater environments (e.g., 1st and 2nd order streams), low flow responses can also include widespread hypoxic conditions and accumulation of reduced inorganic solutes [*Gomez-Gener et al., 2020*], which likely have more powerful influences on aquatic organisms and community structure [*Pardo and Garcia, 2016*]”. (Lines 149-152)

18. Line 134: LMW DOC increased linearly compared to what?

Response: We have added “in relation to prolonged dry periods” (Line 178)

19. Line 140: “expressed to the severity” expressed as...? Can you help the reader by a short complement? From your figures and context, I guess drought severity is measured as

number of low flow days, but you state this nowhere (e.g. “Drought severity” used in L66, 86, 89, 104). At some point you also use the phrase “drought magnitude”, “drought duration”. Is this all the same?

Response: Thank you for your observation, we have changed this to “Thus, the changes in both DOC quantity and quality parameters suggest that solutes accumulated during seasonal low flow periods become mobilized during the rewetting period to an extent that is proportional to antecedent drought severity” (Line 183-186) and have been more consistent with the terminology throughout the manuscript.

Response to reviewer #3 comments

This is an interesting and well written paper on the increasing role of drought in regulating DOC concentrations in boreal rivers. I enjoyed reading the paper and the authors have collected a useful dataset. I am less certain on the impact of this paper, and hence its publication in a nature journal, as the authors are testing known results in a new location. I would recommend the paper for publication as is in a lower ranking journal but feel the authors may need to do more to justify Nature Comms.

Response: The authors are thankful for the encouraging comments by the reviewer and have worked on improving the analysis and scope of the study which we believe better conveys the importance of the research. Specifically, we have supported our analysis of DOC concentrations with flux exports which corroborates the seasonal shifts in DOC in the summer and autumn. We have also discussed the ecological and social implications of change in DOC concentrations and quality in streams. Finally, we have clarified the specific comment and have tested the proportion of inorganic nitrogen to total nitrogen. We hope that with these additions, the importance of this research for highlighting the direct and lagged effects of droughts on DOC quantity and quality makes our work more coherent and interesting.

General Comments

1. The hypothesis (L70 onwards) could be restated as ‘we expect boreal rivers to be the same as non-boreal rivers’ which is what was found. I understand the incidence of drought in these landscapes is somewhat novel, and expected to increase, and therefore suggest the authors perform an analysis of how projected climate change could increase DOC flux from these catchments. Essentially, combine their excellent Fig 7 with a projection of drought frequency/severity in the future so we can understand the scale of the issue.

Response: As noted above, we agree that there have been many studies addressing relationships between flow and DOC – but not so many have focused on the influence of drought, even less so in boreal landscapes, and none in northern boreal landscapes using such a robust, multisite dataset that provides observations during and post-drought over nearly two decades. Further, the idea that summer drought is playing a much larger role than expected in the long-term seasonality of DOC supply to these northern streams is not something that we already knew, even if some of the mechanistic underpinnings seem similar to what has been found in different biomes (e.g., from semi-arid catchments, e.g., in Romani et al. 2006). In addition, while previous studies have focused on understanding changes

in DOC quantity (e.g., Lewis and Grant, 1979; Hornberger et al., 1994, Tunaley et al., 2018), our work addresses the responses relative to the long-term mean and seasonality of DOC. None of these papers show how such responses are mediated by the landscape structure (e.g., in terms of catchment size and wetland cover) in the way that we can here. Finally, unlike most previous studies, our analysis uses a variety of quality parameters (or proxies) to support the underlying changes in DOC quantity in different landscape types relative to drought.

With regards to the climate projections in the future, while most climate projections in Scandinavia indicate a generally wetter climate in the future [Heinrich and Gobiet, 2012; Sein et al., 2014], these projections are based on satellite data of land surface total water storage [Rodell et al., 2018] that has low spatial and temporal resolution and ambiguous representation at various soil depths [Chen et al., 2021]. As such, hydrological summer droughts have rarely been modeled despite having the most severe events occurring in 2018, 2014, 2006 recorded across Europe. A major complicating factor here has to do with the effects of warming on forest evapotranspiration and the extent to which this response may give rise to seasonal hydrologic drought, even under normal precipitation inputs.

As such, we are not currently in a position to add specific future drought predictions concerning our conceptual model. However, Spinoni et al. [2018] use a combined drought indicator based on high resolution (0.11°) bias-corrected regional downscaling to show an increase in drought frequency over Sweden and Europe. These results as well as the increased projection of extreme events in the future by IPCC [2021] indicate that increased drought frequency and severity are expected for the boreal region in the future. These reference has now been added to the manuscript to support this point (Line “266-275”). Thank you for the comment.

2. Furthermore, the authors have an extensive DOC concentration dataset as well as flow data. Why no calculation of the variation in DOC load or flux between catchments, only concentration?

Response: Thank you for this comment which has now been incorporated into the manuscript as supplementary material showing seasonal changes in export and annual exports (Supplementary Figure S1). We agree that DOC flux is important as has been discussed in many other projects previously, however, our primary focus was on the concentration and quality of DOC as this is of more general interest from an ecological and water quality perspective. Still, we calculated the consequences of export at the catchment outlet and found some intriguing results.

The results indicate that while on an annual basis effects of low flow on DOC are not increasing, the seasonal effects correspond to the changes in concentrations we observe in our study. During the summer, prolonged low flows showed the lowest exports (90% lower than the long-term average) while in the autumn, prolonged low flows showed the highest exports (800% higher than pre-drought values (June averages)). However, the years with high autumn precipitation (2017 and 2012)

also showed high exports of DOC. As such, it is not possible to show the importance of low flows on an annual basis when the hydrograph is dominated by high flow events in the autumn and spring. For this analysis, we use summer as the sum of exports from July-August and autumn as October- November (Figure R4). We discuss this as well in comment 6 (Reviewer 2). The seasonal flux analysis has now been included in the manuscript which lends support to the conceptual drought impact section and the seasonal shifts noted in the drought and post-drought analysis (Supplementary analysis Figure S1, Line 101).

Figure R4 DOC export during the summer, autumn, and yearly

3. Could differences in residence time in the catchment explain some of the variation between small and large catchments found?

Response: This is an interesting comment which forms the basis of our network scaling section that larger catchments indeed have intrinsic properties such as longer water travel time that can influence their ability to respond to changes in events. We have used water residence time information from the study catchment [Sterte *et al.*, 2021] to test this concept (Figure R5). The results support the ideas presented in the manuscript that longer residence time in larger catchments explains the variation in the changes in DOC observed during drought and rewetting. While we find this interesting, we do not include this in our study as it extends beyond the scope.

Figure R5 DOC slope changes of drought and Post-drought in relation to the travel time of catchments

Specific Comments

4. L166 onwards is very interesting as it seems to negate the ‘river as chemostat’ hypothesis.

Response: This is a very interesting comment that puts our results into a wider context of the hydrological system. The river network is often viewed as a chemostat (for example [Creed *et al.*, 2015]) where solutes move from high variability to a near steady-state as the order of the stream increases. This has been discussed to be related to variability to terrestrial source areas and the increased disconnection as the stream order increases. This concept explains the variability of stream chemistry in streams in catchments of heterogeneous landscape types and sizes.

However, our analysis which focuses on the responses of the stream chemistry to drought events in catchments of different landscape types and sizes shows larger responses to events as the stream order or catchment size increases. Although the stream chemistry still behaves as a chemostat the drought responses are the opposite. For instance, we still see high concentrations and variability in the headwater compared to the higher-order streams, however their responses to the events, ie, the amount of DOC that decreases/increases relative to the individual catchment long-term averages, increases in relation to the catchment area. As such, it is difficult to negate the “river as a chemostat” in this sense.

5. L191 suggests peat catchments are more resilient to droughts (higher water storage capacity and lower evaporative losses), and therefore the catchments analyzed did not experience the same drought severity in terms of e.g. WT depth change. Does this not affect the analysis across catchments if we’re effectively seeing a milder drought in some?

Response: This is a great comment and we do see an effect across the catchments in how they respond to drought based on their landscape peat coverage.

Since the study catchments are nested, it is not possible to test the effects of an isolated landscape feature without the confounding effects. As such, we expect that the responses were seen at the outlet of the catchment for instance (C16) maybe also include some of the buffering effects of the wetlands found in (C4). However, modeling the responses in the catchments to the areal coverage and peat percentage to find the best explanatory variable across the catchments, the areal coverage was more significant than the landscape types (Figure R6).

This has now been included in the manuscript (lines 238-253) which has improved the cohesion of this section.

Figure R6 landscape model of DOC slope responses

- Inorganic N is <9% of total N and is discarded due to limited data at some sites. Could you perhaps justify this further by showing for the sites there are data that this does not vary with drought or season to give us more confidence this does not explain some of the TN effects?

Response: Based on the reviewer's comment, we have tested if the proportion of inorganic N (DIN) to total N can explain the changes in TN during droughts and

post-drought. From the monthly distribution of DIN to TN, it can be observed that the lowest proportion of inorganic nitrogen occurred during the summer (red bar) when drought was highest across the majority of the catchment with a complete dataset (Figure R7). We expected that if the proportion of DIN can explain the changes we observed in the C/N ratio, then a higher % of inorganic nitrogen would correlate with a larger change in the C/N ratio. Instead, we observed that the sites with the highest proportion of DIN during the summer were the sites with the lowest C/N ratio change in response to drought - suggesting that the contribution of DIN to the TN pools is not driving the changes in C/N observed across the catchments (Figure R8).

For clarity, we have made a note of this in the methods section (lines 336-346).

Figure R7 The proportion of inorganic N to total N during the summer months in the catchments with full datasets

Figure R8 The DOC drought response in relation to the proportion of inorganic N across the catchments in 2018

7. Overall, this is a good paper but I feel to be high impact the authors may have to do a bit more as the results are largely expected.

Response: Thank you for the comment, we hope now that with the suggested changes we have elevated the importance and scope of this research

References

- Acuna, V., I. Munoz, A. Giorgi, M. Omella, F. Sabater, and S. Sabater (2005), Drought and postdrought recovery cycles in an intermittent Mediterranean stream: structural and functional aspects, *J N Am Benthol Soc*, 24(4), 919-933, doi:10.1899/04-078.1.
- Agren, A., I. Buffam, M. Berggren, K. Bishop, M. Jansson, and H. Laudon (2008), Dissolved organic carbon characteristics in boreal streams in a forest-wetland gradient during the transition between winter and summer, *J Geophys Res-Biogeophys*, 113(G3), doi:10.1029/2007jg000674.
- Berggren, M., H. Laudon, and M. Jansson (2007), Landscape regulation of bacterial growth efficiency in boreal freshwaters, *Global Biogeochem Cy*, 21(4), doi:10.1029/2006gb002844.
- Berggren, M., H. Laudon, and M. Jansson (2010), Bacterial utilization of imported organic material in three small nested humic lakes, *Int Ver Theor Angew*, 30, 1393-+.
- Borken, W., and E. Matzner (2009), Reappraisal of drying and wetting effects on C and N mineralization and fluxes in soils, *Global Change Biol*, 15(4), 808-824, doi:10.1111/j.1365-2486.2008.01681.x.
- Bradshaw, C. J. A., and I. G. Warkentin (2015), Global estimates of boreal forest carbon stocks and flux, *Global Planet Change*, 128, 24-30, doi:10.1016/j.gloplacha.2015.02.004.
- Buffam, I., H. Laudon, J. Temnerud, C. M. Morth, and K. Bishop (2007), Landscape-scale variability of acidity and dissolved organic carbon during spring flood in a boreal stream network, *J Geophys Res-Biogeophys*, 112(G1), doi:10.1029/2006jg000218.
- Chen, D. L., P. Zhang, K. Seftigen, T. H. Ou, M. Giese, and R. Barthel (2021), Hydroclimate changes over Sweden in the twentieth and twenty-first centuries: a millennium perspective, *Geogr Ann A*, 103(2), 103-131, doi:10.1080/04353676.2020.1841410.
- Clark, J. M., A. Heinemeyer, P. Martin, and S. H. Bottrell (2012), Processes controlling DOC in pore water during simulated drought cycles in six different UK peats, *Biogeochemistry*, 109(1-3), 253-270, doi:10.1007/s10533-011-9624-9.
- Creed, I. F., et al. (2018), Global change-driven effects on dissolved organic matter composition: Implications for food webs of northern lakes, *Global Change Biol*, 24(8), 3692-3714, doi:10.1111/gcb.14129.
- Creed, I. F., et al. (2015), The river as a chemostat: fresh perspectives on dissolved organic matter flowing down the river continuum, *Can J Fish Aquat Sci*, 72(8), 1272-1285, doi:10.1139/cjfas-2014-0400.
- Dahlen, J., S. Bertilsson, and C. Pettersson (1996), Effects of UV-A irradiation on dissolved organic matter in humic surface waters, *Environ Int*, 22(5), 501-506, doi:10.1016/0160-4120(96)00038-4.
- Fenner, N., and C. Freeman (2011), Drought-induced carbon loss in peatlands, *Nat Geosci*, 4(12), 895-900, doi:10.1038/Ngeo1323.
- Ghil, M., et al. (2011), Extreme events: dynamics, statistics and prediction, *Nonlinear Proc Geoph*, 18(3), 295-350, doi:10.5194/npg-18-295-2011.
- Gomez-Gener, L., A. Lupon, H. Laudon, and R. A. Sponseller (2020), Drought alters the biogeochemistry of boreal stream networks, *Nat Commun*, 11(1), 1795, doi:10.1038/s41467-020-15496-2.
- Granados, V., C. Gutierrez-Canovas, R. Arias-Real, B. Obrador, A. Harjung, and A. Butturini (2020), The interruption of longitudinal hydrological connectivity causes delayed responses in dissolved organic matter, *Sci Total Environ*, 713, doi:10.1016/j.scitotenv.2020.136619.
- Heinrich, G., and A. Gobiet (2012), The future of dry and wet spells in Europe: a comprehensive study based on the ENSEMBLES regional climate models, *Int J Climatol*, 32(13), 1951-1970, doi:10.1002/joc.2421.
- IPCC (2021), Climate Change 2021: The Physical Science Basis. Contribution of Working Group I to the Sixth Assessment Report of the Intergovernmental Panel on Climate Change, Cambridge University Press.

Kalbitz, K., S. Solinger, J. H. Park, B. Michalzik, and E. Matzner (2000), Controls on the dynamics of dissolved organic matter in soils: A review, *Soil Sci*, 165(4), 277-304, doi:Doi 10.1097/00010694-200004000-00001.

Kroer, N. (1993), Bacterial-Growth Efficiency on Natural Dissolved Organic-Matter, *Limnol Oceanogr*, 38(6), 1282-1290, doi:DOI 10.4319/lo.1993.38.6.1282.

Kuhry, P., and D. H. Vitt (1996), Fossil carbon/nitrogen ratios as a measure of peat decomposition, *Ecology*, 77(1), 271-275, doi:10.2307/2265676.

Marty, C., D. Houle, C. Gagnon, and F. Courchesne (2017), The relationships of soil total nitrogen concentrations, pools and C:N ratios with climate, vegetation types and nitrate deposition in temperate and boreal forests of eastern Canada, *Catena*, 152, 163-172, doi:10.1016/j.catena.2017.01.014.

Moatar, F., B. W. Abbott, C. Minaudo, F. Curie, and G. Pinay (2017), Elemental properties, hydrology, and biology interact to shape concentration-discharge curves for carbon, nutrients, sediment, and major ions, *Water Resour Res*, 53(2), 1270-1287, doi:10.1002/2016wr019635.

Muhr, J., J. Franke, and W. Borken (2010), Drying-rewetting events reduce C and N losses from a Norway spruce forest floor, *Soil Biol Biochem*, 42(8), 1303-1312, doi:10.1016/j.soilbio.2010.03.024.

Pacific, V. J., K. G. Jencso, and B. L. McGlynn (2010), Variable flushing mechanisms and landscape structure control stream DOC export during snowmelt in a set of nested catchments, *Biogeochemistry*, 99(1-3), 193-211, doi:10.1007/s10533-009-9401-1.

Pardo, I., and L. Garcia (2016), Water abstraction in small lowland streams: Unforeseen hypoxia and anoxia effects, *Sci Total Environ*, 568, 226-235, doi:10.1016/j.scitotenv.2016.05.218.

Pastor, J., J. Solin, S. D. Bridgham, K. Updegraff, C. Harth, P. Weishampel, and B. Dewey (2003), Global warming and the export of dissolved organic carbon from boreal peatlands, *Oikos*, 100(2), 380-386, doi:10.1034/j.1600-0706.2003.11774.x.

Rodell, M., J. S. Famiglietti, D. N. Wiese, J. T. Reager, H. K. Beaudoin, F. W. Landerer, and M. H. Lo (2018), Emerging trends in global freshwater availability, *Nature*, 557(7707), 650+, doi:10.1038/s41586-018-0123-1.

Rumpel, C., and I. Kogel-Knabner (2011), Deep soil organic matter-a key but poorly understood component of terrestrial C cycle, *Plant Soil*, 338(1-2), 143-158, doi:10.1007/s11104-010-0391-5.

Sein, D. V., N. V. Koldunov, J. G. Pinto, and W. Cabos (2014), Sensitivity of simulated regional Arctic climate to the choice of coupled model domain, *Tellus A*, 66, doi:10.3402/tellusa.v66.23966.

Spinoni, J., J. V. Vogt, G. Naumann, P. Barbosa, and A. Dosio (2018), Will drought events become more frequent and severe in Europe?, *Int J Climatol*, 38(4), 1718-1736, doi:10.1002/joc.5291.

Sterte, E. J., F. Lidman, E. Lindborg, Y. Sjöberg, and H. Laudon (2021), How catchment characteristics influence hydrological pathways and travel times in a boreal landscape, *Hydrol Earth Syst Sc*, 25(4), 2133-2158, doi:10.5194/hess-25-2133-2021.

Tiwari, T., R. A. Sponseller, and H. Laudon (2018), Extreme Climate Effects on Dissolved Organic Carbon Concentrations During Snowmelt, *J Geophys Res-Biogeo*, 123(4), 1277-1288, doi:10.1002/2017jg004272.

Tiwari, T., R. A. Sponseller, and H. Laudon (2019), Contrasting responses in dissolved organic carbon to extreme climate events from adjacent boreal landscapes in Northern Sweden, *Environ Res Lett*, 14(8), doi:10.1088/1748-9326/ab23d4.

Toberman, H., C. D. Evans, C. Freeman, N. Fenner, M. White, B. A. Emmett, and R. R. E. Artz (2008), Summer drought effects upon soil and litter extracellular phenol oxidase activity and soluble carbon release in an upland Calluna heathland, *Soil Biol Biochem*, 40(6), 1519-1532, doi:10.1016/j.soilbio.2008.01.004.

Weishaar, J. L., G. R. Aiken, B. A. Bergamaschi, M. S. Fram, R. Fujii, and K. Mopper (2003), Evaluation of specific ultraviolet absorbance as an indicator of the chemical composition and reactivity of dissolved organic carbon, *Environ Sci Technol*, 37(20), 4702-4708, doi:10.1021/es030360x.

Werner, B. J., A. Musolff, O. J. Lechtenfeld, G. H. de Rooij, M. R. Oosterwoud, and J. H. Fleckenstein (2019), High-frequency measurements explain quantity and quality of dissolved organic carbon mobilization in a headwater catchment, *Biogeosciences*, 16(22), 4497-4516, doi:10.5194/bg-16-4497-2019.

Zarnetske, J. P., M. Bouda, B. W. Abbott, J. Saiers, and P. A. Raymond (2018), Generality of Hydrologic Transport Limitation of Watershed Organic Carbon Flux Across Ecoregions of the United States, *Geophys Res Lett*, 45(21), 11702-11711, doi:10.1029/2018gl080005.

REVIEWERS' COMMENTS

Reviewer #1 (Remarks to the Author):

The authors sent a revised version of their manuscript, which I enjoyed reading. I appreciate the efforts made by the authors in addressing all my comments. The manuscript now includes more absorbance data and the analysis of SUVA values, which I think is very interesting. Moreover, the authors have tested if the normalization of DOC concentrations to discharge would affect their results, which is not the case. Here, they could consider putting figure R1 in the supplementary material to back up their statements made in the text concerning normalization.

In my opinion, the manuscript has improved a lot and I think it is suitable for publication after addressing some minor comments and suggestions.

L70: I suggest writing „carbon/nitrogen (C/N) ratio...”

L72: I suggest writing “As the ratio of absorbance [...] tends to be [...], we used it as a proxy for LMW DOC. Also, is “tend to” the right wording here? Is it an assumption or a fact?

L165-L167: In the text, it is not entirely clear why periods of low flow conditions would lead to an increased production. Maybe add a small explanation and split the sentences into two: one about the increased production and one about the accumulation (as a result of limited export, I suppose). These are two different things in my view.

L204: I suggest you state one more time to which changes the numbers in the brackets are referring, e.g. “increase of ca. 0.2...”

L323: summer (lower case)

L336: I think you can use the abbreviation LMW DOC here, which you have introduced one sentence earlier.

L348: I suppose you mean m-1.

Caption Fig 1.: Are you referring to annual mean discharge values? Please specify.

Caption Fig 2.: SUVA₂₅₄ instead of SUVA.

Fig. 2 and 3: In these figures, the colors from the lines but especially from the dots (due to the transparency) are not easy to link to the legend. I understand that these are a lot of categories but is there the possibility to use another color scale, which would help to distinguish the colors?

L668: I think that a comma, semicolon or full stop before hence is missing.

Fig 4: If looking only at the figure, it is not clear what low, medium and high means. Maybe you could add “number of low flow days” below the three categories?

Supplementary Material

L31: “riparian” is referring to the forest sites, right? Please specify as in the figure itself you write “forest”.

L36: Be consistent with using upper or lower case letters for “drought” and “post-drought”. In L30 you use lower case letters.

Reviewer #3 (Remarks to the Author):

The authors have addressed my comments with both changes to the manuscript, additional analyses and some supplementary figures. I feel this additional work is enough for me to now recommend publication.

REVIEWERS' COMMENTS

Reviewer #1

The authors sent a revised version of their manuscript, which I enjoyed reading. I appreciate the efforts made by the authors in addressing all my comments. The manuscript now includes more absorbance data and the analysis of SUVA values, which I think is very interesting. Moreover, the authors have tested if the normalization of DOC concentrations to discharge would affect their results, which is not the case. Here, they could consider putting figure R1 in the supplementary material to back up their statements made in the text concerning normalization. In my opinion, the manuscript has improved a lot and I think it is suitable for publication after addressing some minor comments and suggestions.

Response: We thank this reviewer for their continued support in improving the quality of the manuscript. All necessary suggestions have been accepted specifically we have included the figure to show the normalized DOC plots in the supplementary material and have correct the manuscript based on the specific comments.

Specific Comments

1. L70: I suggest writing carbon/nitrogen (C/N) ratio?

Response: The suggestion has been made (Line 70)

2. L72: I suggest writing “As the ratio of absorbance [???] tends to be [???], we used it as a proxy for LMW DOC. Also, is ???tend to??? the right wording here? Is it an assumption or a fact?

Response: The suggestion has been made and we have changed “tends to” to “has been found to be” so as to improve the conciseness of the sentence. (Line 72:73)

3. L165-L167: In the text, it is not entirely clear why periods of low flow conditions would lead to an increased production. Maybe add a small explanation and split the sentences into two: one about the increased production and one about the accumulation (as a result of limited export, I suppose). These are two different things in my view.

Response: We have edited this sentence to “Periods of drought that aerate riparian and wetland soils, while also reducing lateral connectivity, can promote DOC production

through decomposition and/or accumulation in upper soil horizons, which can then be mobilized when dry periods are terminated^{15,44}.” (Line 165-167)

4. L204: I suggest you state one more time to which changes the numbers in the brackets are referring, e.g. ???increase of ca. 0.2???????

Response: The suggestion has been made “an increase of” has been added to the brackets. (Line 204)

5. L323: summer (lower case)

Response: This suggestion has been made (Line 322)

6. L336: I think you can use the abbreviation LMW DOC here, which you have introduced one sentence earlier.

Response: This suggestion has been made (Line 335)

7. L348: I suppose you mean m-1.

Response: This suggestion has been made (Line 347)

8. Caption Fig 1.: Are you referring to annual mean discharge values? Please specify.

Response: In this figure we refer to daily mean discharge during the summer period of each year. This has now been added to the figure description, thank you for the comment. (Line 662)

9. Caption Fig 2.: SUVA₂₅₄ instead of SUVA.

Response: This suggestion has been made (Line 668)

10. Fig. 2 and 3: In these figures, the colors from the lines but especially from the dots (due to the transparency) are not easy to link to the legend. I understand that these are a lot of categories but is there the possibility to use another color scale, which would help to distinguish the colors?

Response: Thank you for the comment, we could not find a suitable color scale in r functions which would reduce the visibility problem as they all contain yellow in the spectrum. However, we have opted for increasing the size of the points, regression lines and transparency which has improve the visibility.

11. L668: I think that a comma, semicolon or full stop before hence is missing.

Response: A comma was added based on this suggestion (Line 678)

12. Fig 4: If looking only at the figure, it is not clear what low, medium and high means. Maybe you could add ???number of low flow days??? below the three categories?

Response: This suggestion has been made

Supplementary Material

1. L31: ???riparian??? is referring to the forest sites, right? Please specify as in the figure itself you write ???forest???

Response: Thank you for the comment, we have changed this now to “ forest values” (line 35, 65)

2. L36: Be consistent with using upper or lower case letters for ???drought??? and ???post-drought???. In L30 you use lower case letters.

Response: The D in Drought has been change to lower case here (line 26) and throughout the documents for consistency.

Reviewer #3

The authors have addressed my comments with both changes to the manuscript, additional analyses and some supplementary figures. I feel this additional work is enough for me to now recommend publication.

Response: Thank you for the support